# Ancient environmental DNA reveals shifts in dominant mutualisms during the late Quaternary

Martin Zobel[1], John Davison[1], Mary E. Edwards[2], Christian Brochmann[3], Eric Coissac[4], Pierre Taberlet[4], Eske Willerslev[5,6] & Mari Moora [1]

DNA-based snapshots of ancient vegetation have shown that the composition of high-latitude plant communities changed considerably during the late Quaternary. However, parallel changes in biotic interactions remain largely uninvestigated. Here we show how mutualisms involving plants and heterotrophic organisms varied during the last 50,000 years. During 50–25 ka BP, a cool period featuring stadial-interstadial fluctuations, arbuscular mycorrhizal and non-N-fixing plants predominated. During 25-15 ka BP, a cold, dry interval, the representation of ectomycorrhizal, non-mycorrhizal and facultatively mycorrhizal plants increased, while that of N-fixing plants decreased further. From 15 ka BP, which marks the transition to and establishment of the Holocene interglaciation, representation of arbuscular mycorrhizal plants decreased further, while that of ectomycorrhizal, non-mycorrhizal, N-fixing and wind-pollinated plants increased. These changes in the mutualist trait structure of vegetation may reflect responses to historical environmental conditions that are without current analogue, or biogeographic processes, such as spatial decoupling of mutualist partners.

---

[1] Department of Botany, Institute of Ecology and Earth Sciences, University of Tartu, 40 Lai Street, 51005 Tartu, Estonia. [2] Geography and Environment, University of Southampton, Highfield, Southampton SO17 1BJ, UK. [3] Natural History Museum, University of Oslo, PO Box 1172 Blindern, NO-0318 Oslo, Norway. [4] CNRS, Laboratoire d'Ecologie Alpine (LECA), Univ. Grenoble Alpes, F-38000 Grenoble, France. [5] Centre for GeoGenetics, Natural History Museum of Denmark, University of Copenhagen, Øster Voldgade 5-7, 1350 Copenhagen, Denmark. [6] Department of Zoology, University of Cambridge, Downing St, Cambridge CB2 3EJ, UK. Correspondence and requests for materials should be addressed to M.M. (email: mari.moora@ut.ee)

Biotic interactions, including mutualisms, play an important role in determining species' responses to environmental change and thus mediate community composition and diversity. Through time, both plant life-history traits[1] and biotic interactions are modified by changes in climate[2]. Until now, most attention has been paid to herbivory and specifically its dynamic relationship with vegetation composition.[3,4] Much less is known about the stability, or otherwise, of mutualisms through time. However, the spatial and temporal scales over which such relationships change is an important additional key to understanding ecosystem responses to future climate change[5]. Palaeodata offer important insights into the long-term trajectory of interactions and the longevity of ecosystems[6]. Conceivably insights into key plant mutualisms can be gained from examining floristic composition and related plant traits during the climate changes of the late Quaternary.

Most records of past vegetation derive from fossil pollen, to the extent that records have been collated into comprehensive databases[7,8]. However, many records from north-temperate and boreal regions are dominated by wind-pollinated taxa that are high pollen producers, weighting the signal against insect-pollinated taxa and underestimating species richness in records based on relatively low pollen counts[9]. Macrofossils can provide a greater level of taxonomic resolution than many pollen records, but they are less common, they provide localized snapshots of palaeovegetation and their preservation varies depending upon the durability of the plant tissue[10]. Metabarcoding of ancient DNA is an alternative approach to studying past vegetation[11] that has the potential to produce taxonomically rich records[12]. While it has limitations, such as poor resolution of some taxonomic groups[13] and representational biases among functional groups[14], ancient DNA occurs in most depositional environments, and its analysis can potentially reveal those fractions of plant communities and local floras not reliably retrieved by analysis of pollen or plant macrofossils.

Recently, Willerslev et al.[12] generated an extensive environmental DNA (eDNA) data set describing the floristic composition of vegetation at sites across the largely unglaciated northern high latitudes (Siberia, Alaska and Yukon) during the last 50,000 years. They assigned samples to three time periods, representing a compromise that allowed robust statistical comparisons but also reflected key changes in climatic conditions pertinent to the study region, which differed from those of glaciated Eurasia and North America (15,16; see Methods). Because the period 25 to 15 ka BP was comparatively climatically stable (cold and arid) in the study region, Willerslev et al.[12] designated it the last glacial maximum (LGM). The pre-LGM period (>25 ka BP) was characterized by fluctuating stadial-interstadial conditions[17], while after 15 ka BP (post-LGM), a major climatic transition affected the region, bringing moister and warmer conditions that were a precursor to the establishment of the current interglacial period (18; Holocene; 11.4 ka BP-present). Willerslev et al.'s[12] results indicated the existence of taxonomically rich pre-LGM vegetation that was dominated by forbs and encompassed taxa that typify both tundra and arctic steppe environments. Taxonomic diversity declined during the LGM, and the LGM flora was largely a subset of the pre-LGM flora. Post-LGM diversity increased, without reaching the pre-LGM level, and comprised fewer forbs and more graminoids and trees than in earlier periods, reflecting a shift from dry tundra-steppe to moist tundra and forest. Willerslev et al.[12] linked vegetation structure and composition with megafaunal diet but did not consider other biotic interactions, such as mutualisms. Here, we investigate the representation of mutualisms associated with late-Quaternary vegetation, focusing on three types of mutualistic interaction between plants and heterotrophic organisms: mycorrhizal associations, symbiotic nitrogen fixation, and pollination.

Mycorrhizae are an ancient symbiosis between soil fungi and plant roots. Several types of mycorrhizal symbiosis are recognized, of which three are widespread. Arbuscular mycorrhizae (AM) are the most ancient type and also the commonest, with > 80% of vascular plant species capable of entering into the symbiosis. Ectomycorrhizae (ECM) and ericoid mycorrhizae (ERM) have evolved multiple times, all more recently than AM, and are only formed by approximately 2 and 1% of vascular plant species, respectively. Non-mycorrhizal status (NM), which is also evolutionarily recent, can be attributed to about 6% of plant species[19,20].

With increasing latitude or altitude, predominantly AM vegetation (grassland, forest or shrubland) is replaced by ECM forests with AM understory, then by ECM forests with ERM understory, and finally by ERM dominated heathlands and tundra heaths. One might expect that temporal change in the predominance of mycorrhizal types should follow a broadly analogous pattern to those observed in relation to latitude and altitude, e.g. such that the predominance of mycorrhizal types in historically cooler periods most closely resembles the pattern in contemporary high-latitude vegetation. However, actual data about the occurrence during the late Quaternary of plants with particular mycorrhizal characteristics are scarce and concern differential extinction[21] and migration[22,23] of AM and ECM trees during the Holocene.

If a plant species is involved in mycorrhizal symbiosis, its mycorrhizal status—whether it is obligately mycorrhizal (OM) or facultatively mycorrhizal (FM)—provides an indication of its reliance on the symbiosis[19,24]. In Europe, at the regional scale, OM plant species are known to prefer drier and warmer habitats with higher soil pH, while FM species have broader distributions[25,26]. The benefit of mycorrhizal association may change in relation to $CO_2$ concentration[27], but there is no empirical information about mycorrhizal function in relation to historically lower $CO_2$ concentrations or under conditions of carbon starvation. Nor is there information on how the shares of NM, OM and FM plant species have changed through the late Quaternary.

Symbiotic nitrogen fixation reflects the ability of some plant species to fix gaseous $N_2$ from the air with the help of i) bacteria collectively referred to as *rhizobia*, ii) the actinobacteria genus *Frankia* or iii) cyanobacteria[28,29]. In certain conditions, plants hosting symbiotic nitrogen-fixing microbes are major ecosystem drivers[30,31]. The local abundance of the N-fixing symbiosis depends on temperature, soil resources and other biotic interactions[29], while, at large scales, the rate of biological N-fixation depends on climatic parameters and $CO_2$ concentration[29]. In current vegetation the share of N-fixing plants in communities tends to decline with increasing latitude[32]. However, studies describing variations through geological time in the relative abundance of N-fixing plants are scarce (but see[31]), and we are unaware of any studies relating to the late Quaternary.

Alongside predominantly trophic interactions, such as mycorrhiza and biological N-fixation, plant-pollinator relationships represent an important class of mutualistic interaction. The proportion of animal-pollinated species among Angiosperms varies from 78% in the temperate zone to 94% in the tropics[33]. Empirical information about the geographic distribution of pollination modes remains limited. Some studies have shown that the proportion of particular pollination modes in plant communities may depend on altitude and average wind speed[34,35]. Dalsgaard et al.[36] modelled the impact of historical climate change on current plant-pollinator networks, but we are unaware of similar studies using empirical palaeoecological data. Indeed, it has generally been difficult to assess the relative abundance of different pollination modes in palaeovegetation because wind-

**Table 1 Correlation between mutualist traits and growth form in a taxon list of 131 plant MOTUs recorded in northern high-latitude permafrost samples[12]**

| | Growth form | Mycorrhizal type | Mycorrhizal status | Pollination | N fixation |
|---|---|---|---|---|---|
| Growth form | — | 0.72 | 0.40 | 0.83 | 0.16 |
| Mycorrhizal type | <0.001 | — | 0.79 | 0.35 | 0.25 |
| Mycorrhizal status | <0.001 | <0.001 | — | 0.26 | 0.20 |
| Pollination | <0.001 | 0.002 | 0.02 | — | 0.13 |
| N fixation | 0.31 | 0.04 | 0.07 | 0.17 | — |

Cramer's *V* (chi square scaled from 0 to 1; upper triangle) and *P* (lower triangle) are presented

pollinated species predominate in pollen records. Using DNA-based techniques to estimate the proportions of wind-pollinated plants in late Quaternary vegetation could provide an important insight into potential biases affecting pollen-based analyses.

Here, we integrate available data on plant mutualistic traits with Willerslev et al.'s[12] data set to investigate how the abundance of plants associated with different symbioses changed during the late Quaternary. Contemporary correlates of mutualist trait abundance provide some basis for making general predictions about temporal patterns during the late Quaternary. Since most traits have been shown to vary in relation to latitude we hypothesized that trends in the relative abundance of mutualist types through warmer and cooler periods of the late Quaternary might broadly reflect patterns seen along contemporary latitudinal gradients. However, we note that modern analogues for past vegetation (and associated mutualisms) are incomplete, reflecting the large difference in climatic conditions between glacial and interglacial periods[37,38]. Indeed, our analyses show that changes in the relative abundance of mycorrhizal types and statuses, N-fixing plants and insect-pollinated plants in late-Quaternary vegetation do not consistently reflect patterns seen along contemporary latitudinal gradients. Instead, alternative patterns of mutualistic relationship may have been driven by the direct or interactive effects of non-analogous environmental conditions or by biogeographic processes accompanying climate changes.

## Results

**Trait abundance and correlation**. Among the molecular operational taxonomic units (MOTUs) retained for analysis, 86 (66%) were forbs, 28 (21%) were graminoids, 11 (8%) were dwarf shrubs and 6 (5%) were trees or shrubs; 86 (66%) were AM, 7 (5%) were ECM, 10 (8%) were ERM and 28 (21%) were NM; 58 (44%) were FM and 45 (34%) were OM; 9 (7%) formed N-fixing symbioses and 122 (93%) did not form N-fixing symbioses; 87 (66%) were insect-pollinated and 44 (34%) were wind-pollinated.

Mycorrhizal traits are known to be related to growth form[19], and our data confirmed these correlations (Table 1). For instance, we recorded relatively more trees and shrubs among ECM and ERM species, which in turn tended to be obligately mycorrhizal species. We also recorded varying degrees of correlation between all other traits (Table 1). For instance, there were no NM species among N-fixers in this data set.

**Community-level trait responses to climate**. The growth form composition of vegetation across the study region, as estimated from the proportion of DNA sequence reads representing the different categories, varied between climatic periods (Fig. 1 and Table 2; see ref. [12]). The post-LGM was characterized by fewer forb reads and more reads corresponding to other growth form categories, compared with both of the earlier periods. The proportions of taxa falling into the different growth form categories followed a similar pattern (Fig. 1).

The relative abundance of mycorrhizal types, expressed either as the proportion of sequence reads or the proportion of MOTUs, was significantly related to climatic period (Fig. 1 and Table 2). In both analyses, the relative proportions of mycorrhizal types differed significantly between pre-LGM and post-LGM, while the LGM was intermediate (Table 2). During the pre-LGM, AM host plants predominated, with other mycorrhizal types present at lower abundance. During the LGM, the shares of AM plants decreased while the shares of ECM plants and NM plants increased. In the post-LGM climatic period, the proportions of AM and ERM plants decreased, while the shares of ECM and NM plants increased.

The relative abundance of plants exhibiting different mycorrhizal statuses was also significantly related to climatic period, both when considered as the number of sequence reads or the number of MOTUs (Fig. 1 and Table 2). The proportions of reads corresponding to different mycorrhizal statuses differed between all climatic periods, while the proportion of MOTUs differed significantly between pre-LGM and post-LGM, with the LGM intermediate (Table 2). Pre-LGM was characterized by a relatively high proportion of OM plants, while in later stages the proportions of FM (particularly LGM) and NM (particularly post-LGM) plants increased.

The share of N-fixing plants in vegetation was lower in the LGM than during both pre-LGM and post-LGM (though not significantly different from pre-LGM for the proportion of reads; Fig. 1 and Table 2), while the abundance of different pollination modes did not vary significantly between climatic periods (Fig. 1 and Table 2).

Most trait categories exhibited positive spatial autocorrelation in trait composition at 0–200 km and negative autocorrelation at >2000 km (Supplementary Fig. 1). However, the magnitude of correlation was low and had a minor influence on predicted type I error rates in our analyses (Supplementary Fig. 2 and Supplementary Table 1). The results of most trait–climate models were consistent with a scale-independent trait–climate relationship; though some results relating to growth form and mycorrhizal type were ambiguous (Supplementary Fig. 3).

**Traits associated with MOTU responses to climate change**. We used an ordination approach—the outlying mean index (OMI)—to approximate the position of MOTUs in a climatic niche space defined by the pre-LGM, LGM and post-LGM periods. Niche positioning is calculated as the distance between the mean environmental conditions used by the taxon (its occurrence in different periods) and the mean conditions of the study as a whole (i.e., reflecting the extent to which different periods were sampled[39]). The average OMI of individual MOTUs was significantly greater than expected at random ($P < 0.01$), supporting the positioning of MOTUs on the ordination plot (Fig. 2). When traits were considered in isolation, growth form, mycorrhizal type, mycorrhizal status and pollination mode all

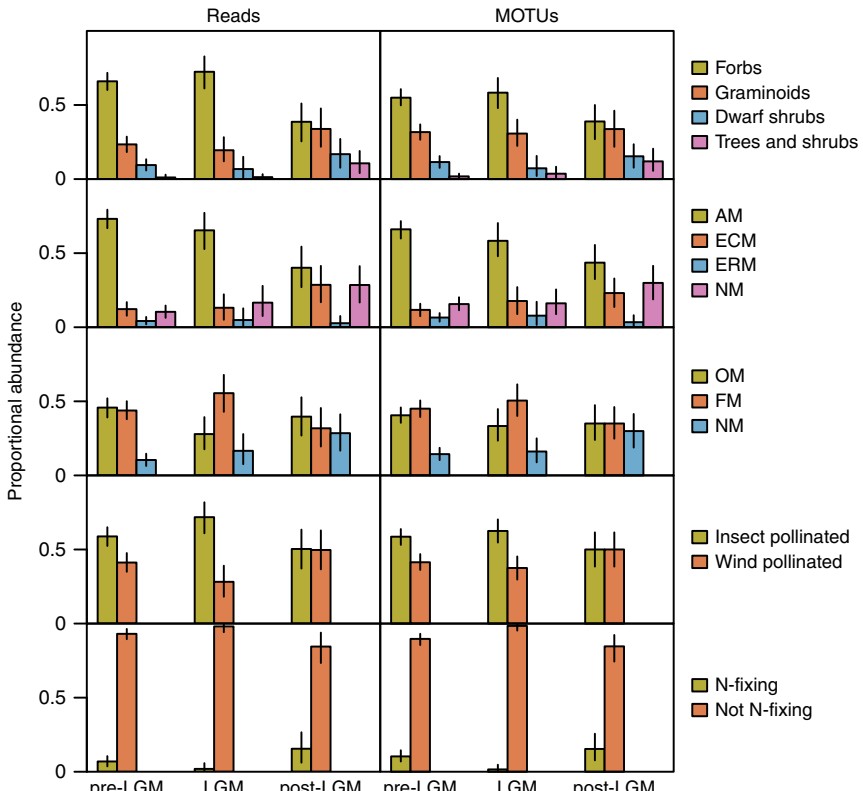

**Fig. 1** Functional structure of past northern high-latitude vegetation. Representation of growth forms, mycorrhizal types, mycorrhizal statuses, pollination types and N-fixing taxa among plants recorded from permafrost samples ($n = 216$) dated to three climatic periods (pre-LGM, $n = 145$; LGM, $n = 32$; post-LGM, $n = 39$) during the late Quaternary. The proportional abundance of DNA sequence reads and MOTUs (molecular operational taxonomic units) is shown. Error bars show bootstrapped 95% confidence intervals

had significant power to explain MOTU niche locations (Table 3). After accounting for the effects of growth form and other non-confounded mutualist traits (mycorrhizal type and mycorrhizal status were strongly correlated and partially confounded, so were not included in the same model), only mycorrhizal status retained significant explanatory power.

The position of MOTUs differed significantly between trait categories along the first OMI axis (pre-LGM/LGM vs post-LGM) but not along the second (pre-LGM/post-LGM vs LGM) (Fig. 3). The average niche position of forb MOTUs was located significantly closer to the pre-LGM/LGM and further from the post-LGM (axis 1) than all other growth forms. The average niche position of AM MOTUs mirrored that of forbs, being located significantly closer to the pre-LGM/LGM and further from the post-LGM (axis 1) than all other mycorrhizal types. FM MOTUs were also located closer to the pre-LGM/LGM (axis 1), and NM closer to the post-LGM (axis 1), compared with other mycorrhizal statuses. MOTUs belonging to different N-fixing categories did not differ in average niche position along either axis. Wind-pollinated MOTUs were located closer to the post-LGM and insect-pollinated MOTUs closer to the pre-LGM/LGM.

The estimated positioning of different mutualist trait categories in climatic niche space was relatively similar in non-phylogenetic models and in models where phylogenetic signal was co-estimated with other model parameters (Supplementary Fig. 4). Nonetheless, mean positions along axis 1 were generally estimated with less precision by the phylogenetic model, and all categories of mycorrhizal status, N-fixing status and pollination mode showed relatively greater affiliation with the post-LGM in the phylogenetic model.

## Discussion

The distribution of mutualist traits in northern high-latitude vegetation has fluctuated during the last 50,000 years. The relatively taxon-rich vegetation of the pre-LGM was dominated by taxa that form AM, with fairly equal abundances of OM and FM taxa. Most taxa were non-N-fixing during this period. During the LGM, plant species richness decreased[12], and the mutualist characteristics of the vegetation changed: the representation of AM taxa decreased, while that of ectomycorrhizal (ECM) and non-mycorrhizal (NM) taxa increased, as did the share of FM taxa. The proportion of N-fixing taxa decreased. Plant species richness increased again in the post-LGM, though it did not reach the level of the pre-LGM[12]. Notably, there was a further decrease in the share of AM taxa and an increase of ECM and NM taxa. The share of N-fixing taxa increased to approximately pre-LGM levels, though there was no association between the N-fixing trait and MOTU climatic niche positioning. Temporal variations in the pollination trait also received inconsistent statistical support: temporal patterns in the share of pollination types were weak, but wind-pollinated MOTUs were significantly associated with the post-LGM. These results indicate weak support for some contemporary environmental correlates as drivers of mutualist change through the late Quaternary, but no support for others. Our hypothesis that trends in the abundance of mutualisms during cool and warm periods of the late Quaternary would mirror patterns seen along contemporary latitudinal gradients was generally not supported.

Both physical data[40], data-model comparisons[41] and ecosystem function reconstructions taking into account the extinction of megafauna[4] attest that conditions in the study region for much of the late Quaternary do not have direct analogues today[16,37,38,42].

**Table 2 Variation in the relative abundance of sequencing reads corresponding to different trait categories (Reads) and in the relative abundance of taxa in different trait categories (MOTUs) in relation to climatic period (pre-LGM, LGM, post-LGM)**

| Trait response | Reads | | | | | MOTUs | | | | |
|---|---|---|---|---|---|---|---|---|---|---|
| | R2 | P | pre-LGM | LGM | post-LGM | R2 | P | pre-LGM | LGM | post-LGM |
| Growth form | 0.05 | 0.01 | a | a | b | 0.02 | 0.05 | a | ab | b |
| Mycorrhizal type | 0.07 | 0.001 | a | ab | b | 0.04 | 0.004 | a | ab | b |
| Mycorrhizal status | 0.04 | 0.01 | a | b | c | 0.03 | 0.04 | a | ab | b |
| Pollination | 0.02 | 0.07 | — | — | — | 0.01 | 0.24 | — | — | — |
| N fixation | 0.03 | 0.04 | a | a | b | 0.03 | 0.05 | a | b | a |

R2 and P value from PERMANOVA are presented. Where a significant difference was detected among climatic periods, different letters are used to distinguish climatic periods that differed significantly in pairwise PERMANOVA analysis.

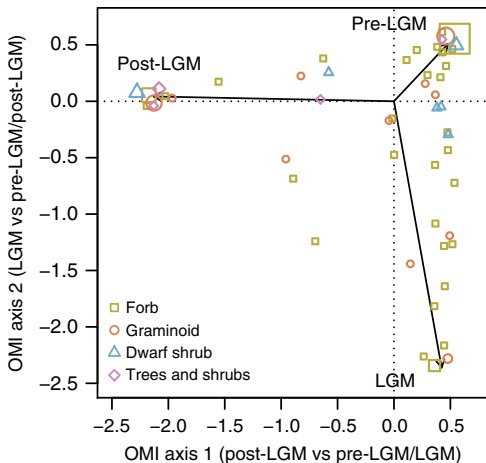

**Fig. 2** Positioning of northern high-latitude plants in historical climatic niche space. The OMI constrained ordination approach is used to estimate the niche position of plant species (points) as defined by their abundance in three climatically different periods during the late Quaternary: pre-LGM, LGM and post-LGM. Symbol sizes are scaled to represent the density of data points at identical positions in ordination space

Below we consider how such non-analogous aspects of the climate and environment may have contributed to the results we observed. We also consider whether biogeographic processes and biotic interactions within the wider ecosystem might have played a role. While we cannot explicitly address the mechanisms underlying the observed patterns on the basis of descriptive data, we propose what we believe to be plausible explanations.

Trends in the abundance of mycorrhizal types across warmer and colder periods of the late Quaternary did not mirror patterns seen along contemporary latitudinal and elevational gradients. We observed a shift from AM to ECM and NM in the post-LGM, but, interestingly, not to ERM, even though Ericales are most dominant in post-LGM pollen records[43]. Trends in precipitation may be an important driver of these changes. High soil moisture and low temperature are associated with NM status and low representation of AM species in contemporary northern-hemisphere vegetation[25,26,44]. Increasing precipitation across the study region from ca. 14 ka BP onwards[16,45,46] resulted in higher soil moisture, making the post-LGM the wettest of the three periods. This shift is therefore consistent with the observed NM increase and AM decrease in the post-LGM. The changing level of atmospheric $CO_2$ may also have played a role, given the strong increase in $CO_2$ concentrations from as low as 190 ppm during the LGM to ambient post-industrial levels of 300–400 ppm[47]. The efficiency of AM associations has been shown to decrease with

increasing $CO_2$ (34), while ECM plants exhibit a strong positive response to elevated $CO_2$[27].

Plant associations with mycorrhizal fungi and N-fixing microorganisms may also have been influenced by variation in the availability of potentially limiting nutrients. Nitrogen is thought to be the most limiting nutrient for plant productivity in many contemporary northern ecosystems, although co-limitation by P is not uncommon[48–50]. In general, AM particularly enhances plant P nutrition, while ECM allows plants to obtain N[19]. On this basis, the observed relative abundances of these mycorrhizal types suggest that P-limitation in pre-LGM vegetation may have been replaced by N-limitation in post-LGM vegetation. Increasing precipitation with the onset of interglacial conditions led to the development of more organic soils and to paludification in some parts of the study area[51]. These changes are likely to have offset, at least in part, increases in mineralization of both N and P with rising temperatures[52], although the effect may have been dependent on vegetation type[53]. Aerts et al.[49] showed experimentally that moderate changes in summer temperatures stimulate litter P release while having limited effects on litter N release in sub-arctic ecosystems. As a result, N-limitation of plant growth in such ecosystems may be promoted by warming. While no direct evidence of a transition from P-limitation to N-limitation during the last 50 ka exists, a global comparison of sedimentary nitrogen isotopes and carbon accumulation across the glacial–deglacial transition led McLauchlan et al.[54] to conclude that terrestrial nitrogen availability generally declined as global climates warmed at the onset of interglacial conditions.

Because the magnitude of biological nitrogen fixation is positively related to ecosystem evapotranspiration[16], and considering that $CO_2$ levels and soil C:N ratios were lowest in the LGM[54], the share of N-fixing plants might be expected to be moderate during the pre-LGM, low during the LGM and highest during the post-LGM. The recorded fluctuation in the share of N-fixing plant taxa was partially consistent with this because the total share of N-fixing plants was slightly higher during the post-LGM.

Warmer regions of the globe tend to harbour higher proportions of insect-pollinated plants[55], so it might be expected that warmer climatic periods have also been associated with higher levels of insect pollination. However, the contemporary latitudinal gradient incorporates a larger temperature range than the temporal gradient of the late Quaternary. Indeed, we found that the proportion of insect-pollinated plants did not vary between climatic periods, and ordination analysis suggested that, if anything, wind pollination was more associated with the post-LGM. However, pollination traits were not strongly associated with plant climatic niche positioning once growth form differences were accounted for. Analogous results were recorded by Moeller et al.[56] in relation to a contemporary latitudinal pattern. Indeed, many ectomycorrhizal trees and non-mycorrhizal sedges, which increased during the post-LGM, are wind pollinated. The

**Table 3 Power of mutualist traits and growth form to predict the climatic niche of 131 plant MOTUs in northern high-atitude permafrost samples[15]**

|  | Univariate | After growth form | After growth form and non-confounded traits |
|---|---|---|---|
| Growth form | 0.09** | — | — |
| Mycorrhizal type | 0.08** | 0.03 | 0.03 |
| Mycorrhizal status | 0.04** | 0.05* | 0.05* |
| Pollination | 0.04** | <0.01 | <0.01 |
| N fixation | <0.01 | <0.01 | <0.01 |

MOTU climatic niche was defined as the species scores from a constrained ordination (outlying mean index; OMI) of MOTU relative abundance in relation to a three-level factor describing the climatic period (pre-LGM, LGM and post-LGM). The effects of different trait categories were tested using PERMANOVA in a series of models: (i) univariate—each trait was entered into models as the sole predictor variable; (ii) after growth form—the effect of each trait was tested after accounting for the variation explained by growth form; (iii) after growth form and non-confounded traits—the effect of each trait was tested after accounting for growth form and non-confounded mutualist traits. Mycorrhizal trait and mycorrhizal status were not included in the same models since they were partially confounded. The $R2$ values for individual variables are presented. **$P > 0.01$ *$P > 0.05$.

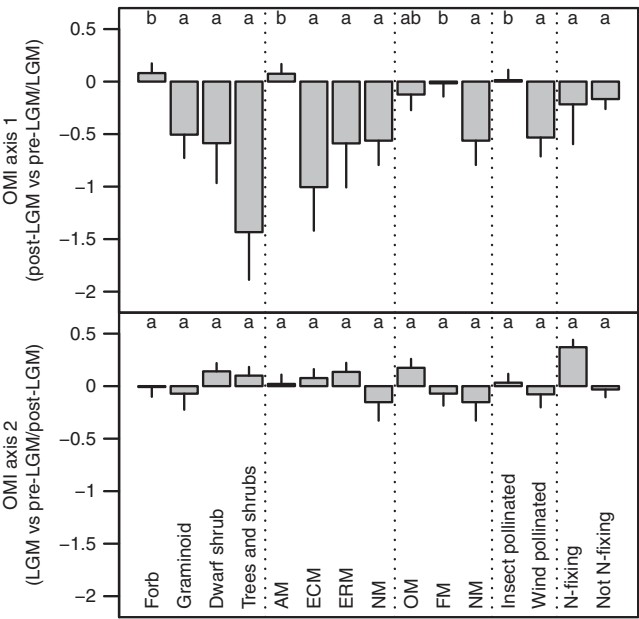

**Fig. 3** Outlying mean index (OMI) axis scores of plants with different mutualist traits. OMI axis scores represent an estimate of species' niche positioning. The first axis corresponds to the distinction between post-LGM (more negative values) and earlier periods (more positive values), while the second corresponds to the distinction between LGM (more negative values) and other periods (more positive values). The mean value (±SE) for each trait category is presented

tendency for wind pollination to increase in the post-LGM may indicate that pollen-based analyses capture a larger fraction of existing diversity in this period compared with earlier periods.

Besides important changes in the abiotic environment, the late Quaternary has been characterized by dynamic patterns of organism distribution, including individualistic species responses to climate change[37] and the extinction of certain mega-herbivores[4]. The onset of the post-LGM represents a period when temperate organisms recolonized higher latitudes and boreal floras occupied many areas previously dominated by arctic floras. This spatial reorganization was reflected in high temporal rates of compositional change in plant communities in eastern North America[37,57] and Europe[58]. The post-LGM vegetation communities investigated here also exhibited high spatial turnover and differed from those that had been locally present during either the pre-LGM or the LGM[12]. Among the biological communities that (re)colonized local sites as the climate warmed during the post-

LGM, some potential mutualistic partnerships may not have re-formed if the taxa involved possessed differing dispersal abilities. Analogous disconnects between potential partners can occur during primary succession when plants and AM fungi colonize new sites[59,60], but it is unclear to what extent such processes occur over much larger temporal and spatial scales[22,23]. Nonetheless, the high proportions of NM and FM taxa in the post-LGM are consistent with this scenario.

Major vegetation changes marking the transition to interglacial conditions were broadly coincident with the extinction of megaherbivores in northern Eurasia and North America[4,42]. Changes in the mutualist trait structure of vegetation during the post-LGM—such as the increase of ECM woody plants and decrease of predominantly AM forbs—may have been to some extent related to a reduction in herbivory and disturbance. Nonetheless, it is notable that mycorrhizal status remained a significant predictor of species responses to climatic change even after accounting for the effect of growth form.

Incorporation of phylogeny into models decreased the precision of estimated trait–environment relationships. Where phylogenetic and non-phylogenetic models suggest similar relationships but with stronger effects in the non-phylogenetic model, this may indicate that trait and niche are to some degree phylogenetically conserved, but that some differentiation is apparent within recently emerged clades[61]. If similarly conserved but unmeasured traits were critical to plant survival under late-Quaternary conditions, such traits might also underlie the patterns observed here. However, we note that mycorrhizal traits—specifically mycorrhizal status—were significant predictors of taxon responses to climate change, even when the effects of other mutualist traits and growth form were accounted for. Therefore, we consider the evidence for a direct link between mycorrhizal traits and taxon responses to be the strongest to emerge from these analyses, although responses associated with other traits should not be discounted.

Our results make the implicit assumption of scale-independent correlation between environment and traits, i.e., a consistent relationship throughout the study region. To the degree that we were able to assess this, the data were largely consistent with this assumption. However, it should be acknowledged that the available sample set provided spatially and temporally uneven coverage[12], which, allied with the moderate total sample size, meant that we did not have high power to detect short-term or regional variations. We therefore do not dismiss the possibility that mutualisms across different regional biotas within the study area responded differently to the environmental changes of the last 50,000 years.

Many forecasts of the impact of climate change on biodiversity are made using macroecological or species distribution models[62] that incorporate information about current relationships between

organisms and the abiotic environment, usually climate parameters. The results of this study show that temporal changes in the abiotic environment were paralleled not only by shifts in floristic composition but by changes in the structure of mutualist relationships among late-Quaternary plant communities. In general the patterns of mutualist trait abundance seen along contemporary latitudinal gradients did not make good predictors of variations in trait abundance between the cooler and warmer periods of the late Quaternary. We have attempted to provide plausible ad hoc explanations for the observed patterns. The complexity of observed responses, even based on a relatively small suite of traits, indicates how features not directly related to climate may be important in predicting trait distributions, and hence floristic composition, in communities.

## Methods

**Ancient DNA data set.** We used data from Willerslev et al.[12], who identified plant species or species groups (molecular operational taxonomic units; MOTUs) from 242 permafrost sediment samples collected from 21 sites in the northern high latitudes. Of these, 221 samples came from northern Siberia, Alaska and northwest Canada, regions left largely ice-free during the last glacial maximum that preserve older deposits. Sample radiocarbon ages ranged from modern to approximately 50 ka BP[63]. Some of the oldest sample ages (40–50 ka BP) may actually be infinite, though most samples are stratigraphically consistent with an MIS-3 designation.

Samples were assigned to a simplified temporal classification relevant to the climate history of the study region: >25 ka BP, 25 to 15 ka BP, <15 ka BP, and given the labels pre-LGM, LGM and post-LGM for ease of reference. Samples dating to the pre-LGM and LGM were largely derived from yedoma, ice-rich silt deposits that accumulated, mostly as a result of nutrient-rich loess deposition[40], under cold, dry climates. The sampled deposits were incorporated into the permafrost zone after deposition, but each 100-mm diameter sample represents years of accumulation and may have been subject to bioturbation/cryoturbation over a few centuries prior to freezing, depending upon active-layer depth and sediment accumulation rate (e.g., given a sedimentation rate of 1 mm per yr and active-layer depth 300 mm: a sample would have accumulated over 100 yr and seasonally thawed for 300 yr). Samples were collected away from the contacts between ice wedges and sediment to ensure that material that may have been moved vertically in relation to frost-cracking and ice-wedge growth was not sampled. Although sites are widely separated, from their current landscape position, many localities were sited on low-lying interfluves meaning that the sampled landscape was more homogenous than if all landscape units (e.g., river valleys, mountaintops) were sampled evenly. These landscapes supported a diverse Pleistocene megafauna[64]. Post-LGM samples were more varied: some were from yedoma, others from lake sediment, peat and soil. This reflects a major rearrangement of northern landscapes and soils, which featured cessation of loess deposition, development of wetlands and organic soils (with a concomitant reduction in nutrient availability), the northward spread of boreal forest into previous treeless vegetation, and the extinction of many megafaunal taxa. While their origins are more varied, the samples nevertheless record the consequences for vegetation composition of this major environmental shift.

Willerslev et al.[12] sequenced the P6 loop of the chloroplast *trn*L (UAA) intron, generating 14,601,839 reads after stringent quality filtering. Reads were assigned to MOTUs present in Arctic and boreal *trn*L taxonomic libraries[13] or GenBank on the basis of sequence similarity (using the programme *ecotag*[65]). Rare MOTUs (<5 reads) were omitted[12]. In total a match was found for 7,738,725 reads and these were presented in Supplementary Table 2 of Willerslev et al.[12].

Because the relative number of reads of a given taxon has been shown to be a sufficiently good proxy of its relative cover or biomass in a vegetation sample[14,66], Willerslev et al.[12] used the quantitative information in their data as a proxy for MOTU abundance. We use the same time period classification and quantitative data in our analysis.

**Plant traits.** Willerslev et al.[12] used information in the BiolFlor database[67] to assign MOTUs to the following growth form categories: forbs, graminoids (grasses, sedges and rushes), dwarf shrubs or other woody plants (trees and shrubs). Where MOTUs could only be linked with species groups or higher-level taxa, and all constituent species had the same growth form, this was allocated to the MOTU; otherwise the MOTU was left 'undefined'. Growth form was allocated to 147 of the 154 identified MOTUs. We used the same approach to compile information on MOTUs for mycorrhizal traits, N-fixation and pollination mode traits from available databases (BiolFlor, Mycoflor[26]) and additional literature searches.

To assign a category of mycorrhizal type (AM, ECM, ERM and NM) and status (FM, OM and NM) to each plant taxon (NM was considered both in the context of types and status), we compiled mycorrhizal information from published data sets, starting with the most recent before moving to progressively older sources—from Hempel et al.[26] to Akhmetzhanova et al.[68], Wang and Qiu[69] and Harley and Harley[70]—as the more recent data sets generally cover, update and correct earlier

information. This process was done carefully to avoid errors propagating in the literature. We performed a review of the mycorrhizal trait data set, limiting possible sources of error as indicated by Brundrett[20] and quality controlling the data as described in Bueno et al.[25] Four and six MOTUs were allocated to the 'undefined' category for mycorrhizal status and type, respectively.

To identify which MOTU component plant species were involved in N-fixing mutualisms, we used data from the BiolFlor database and an additional literature search. MOTUs were assigned to two categories: involved or not involved in N-fixing mutualism. Four MOTUs were undefined.

We categorized MOTUs according to the recorded pollination mode in the BiolFlor database. We considered only two major pollination modes: (1) wind pollination and (2) insect pollination. Although many arctic plant populations are predominantly self-pollinated, mixed-mating (i.e., combining self-pollination with mostly insect-mediated cross-pollination), or partly or fully agamospermous, this may vary within a species across its range[71,72] and there is insufficient information available to classify such species separately in our data set. Because those taxa in our data set that are known to exhibit some degree of self-pollination or agamospermy have insect pollination syndromes (i.e., they are derived from insect-pollinated ancestors), we classified them as insect-pollinated. Nine MOTUs were 'undefined'.

After compiling the trait database, we removed 14 MOTUs (2226214 reads) that could not be defined for all traits and five samples that no longer contained any MOTU data. This left a data table containing 216 samples (145 pre-LGM, 32 LGM and 39 post-LGM), 131 MOTUs (98 pre-LGM, 40 LGM and 51 post-LGM) and 5026792 reads (2469914 pre-LGM, 469290 LGM and 2087588 post-LGM). The precision of taxonomic and trait composition estimates was similarly high in all periods (Supplementary Fig. 5).

**Sensitivity analysis.** Plant trait databases (e.g., Biolflor) almost certainly include errors resulting from misidentification of traits and incorrect transfer of information among sources. The level of error among mycorrhizal trait categorizations has been estimated at 10% of species[20]. We assessed the robustness of our data set to potential errors by introducing random errors into some of the mutualist trait assignments. We assumed that errors may affect assignment of mycorrhizal status, mycorrhizal type (except ERM) and pollination mode; by contrast, we assumed that ERM mycorrhizal type and N-fixing associations, which are phylogenetically conserved, should be reliable. Thus, for each susceptible trait we randomly selected species corresponding to a given error rate (we considered rates between 1 and 20%). For these species, we reassigned the trait category to a randomly selected alternative category. We repeated this process 1000 times at each error rate to generate replicate data sets with a given level of randomly introduced error. We then calculated the proportional abundance of each trait in every sample and correlated these values with the trait abundance observed in the original data. We also recalculated the major analyses described below assuming a 10% error rate in susceptible traits. We found that while trait abundance varied slightly between data sets using original and random trait errors, the trait responses to climatic period remained very similar, with the possible exception of ECM, which was less strongly associated with the post-LGM in data sets incorporating random error (Supplementary Figs. 6 and 7). Also, the results of the most conservative multivariate analysis—the effect of mutualist trait category on species abundance after accounting for growth form and confounded traits—did not change qualitatively in analysis of randomly altered data sets: mycorrhizal status generally remained a significant predictor; original $R2 = 0.04$; original $P = 0.01$; median $R2$ from random analyses = 0.03; median $P$ from random analyses = 0.04.

**Data analysis.** To assess the abundance of different mutualist traits in different climatic periods (pre-LGM, LGM and post-LGM), we calculated for each sample the relative share of each mutualist category in the MOTU list, as well as the relative abundance of MOTUs in each mutualist category, estimated on the basis of read frequency. The shares of different categories among the total pool of reads recovered from a sample resembles what is known as the community weighted mean in plant community ecology[73]. We used bootstrapping to estimate confidence intervals around the shares of different categories: we resampled the raw data table 1000 times, randomly selecting samples with replacement within each climatic period category. The 2.5 and 97.5% percentiles of the bootstrapped distributions were used to estimate 95% confidence intervals. For each trait response matrix we used permutational multivariate analysis of variance (PERMANOVA; function adonis in R package vegan with Bray-Curtis distance[74]) to test the null hypothesis of no difference in trait category distribution (both number of taxa and number of reads per category) between climatic periods.

We also used a constrained ordination approach—the outlying mean index (OMI)—to identify the mutualist traits associated with MOTU-level responses to climatic period. OMI approximates niche positions for species (i.e., MOTUs in this case) by calculating the distance between the mean conditions used by a species and the mean conditions available in a study (i.e., species OMI[39]). Unlike some other constrained ordination techniques, it is applicable to environmental gradients of any length and is not influenced by variations in site richness. A recent review of methods proposed OMI as a stable and sensitive tool for investigating trait–environment relationships[75]. We used OMI to generate a constrained ordination of MOTU abundance in response to climatic period (a factor with

three-levels: pre-LGM, LGM and post-LGM). The species scores from this ordination thus potentially represented MOTU locations in a two-dimensional 'climatic' niche space, with the first axis representing a gradient from pre-LGM and LGM on one hand to post-LGM on the other; and the second axis representing a gradient from pre-LGM and post-LGM on one hand to LGM on the other. We used a Monte Carlo randomization test to test whether average MOTU OMI was greater than would be expected at random (i.e. whether the climatic gradients had significant explanatory power[39]).

Using the species scores on the constrained ordination axes as the multivariate response variable in PERMANOVA (Euclidian distance), we tested the null hypothesis of no difference in MOTU climatic niche location in relation to mutualist trait category. We constructed models where the effects of single explanatory traits (the mutualist traits plus growth form) were (i) tested in isolation, (ii) tested after accounting for growth form (for mutualist traits); and (iii) tested after accounting for growth form and non-confounded mutualist traits (for mutualist traits; since mycorrhizal type and mycorrhizal status were strongly correlated and partially confounded, these were not entered into the same model). To identify the climatic periods associated with particular trait categories, we tested differences between trait categories in the MOTU scores on the separate individual axes. To test the null hypothesis of no difference between categories, we used Dunn's non-parametric test, since the residuals from parametric models were not normally distributed.

Willerslev et al.[12] demonstrated spatial distance decay in the compositional similarity of plant communities in the permafrost data set. Spatial autocorrelation can lead to inflated Type I error in statistical hypothesis tests, while spatial dependence in trait–environment relationships limits the generality of global estimates of trait–environment correlation[76,77]. We assessed the degree of spatial autocorrelation and scale-dependence in trait–climate relationships using multi-scale ordination (MSO)[77] and Mantel correlograms. Multi-scale ordination is a method based on canonical correspondence analysis that decomposes total, explained and residual variance at different spatial scales. A systematic difference between total variance and the sum of residual and explained variance can indicate scale-dependent species-environment correlation. To construct correlograms, we used distance-based redundancy analysis (Bray-Curtis distance), which is similar to PERMANOVA but allows the calculation of a residual matrix for use in the correlogram. The MSO procedure also generates an estimate of the degree to which residual variance is underestimated as a result of autocorrelation. We used these values to estimate the influence of measured autocorrelation on Type I error rates in PERMANOVA.

Analytical approaches that incorporate phylogenetic relationships between taxa can help to link the evolution of functional traits with adaptation to abiotic or biotic environments. Such 'phylogenetic correction' can provide a surrogate of functional trait differences between taxa, and may be particularly important in the presence of missing or complex trait information[61]. We constructed a phylogeny to represent approximate relationships between the plant MOTUs used in this analysis. To do this, we used a backbone supertree of the European flora[78]. We first identified MOTUs that corresponded to single plant species that were present in the backbone phylogeny (n = 26). We then grafted terminal branches onto the tree to represent MOTUs that comprised multiple species or higher taxa. For each such MOTU we attached the terminal branch to a node in the phylogenetic tree representing the lowest taxonomic level among genus (75 MOTUs), tribe (7 MOTUs), subfamily (5 MOTUs) and family (18 MOTUs) that comprised all component taxa of the MOTU and was identifiable in the tree. We then pruned all redundant species in the tree to leave a phylogeny containing 131 MOTU tips (Supplementary Fig. 8). We incorporated this phylogeny into a reanalysis of the relationship between MOTU trait categories and MOTU scores along individual OMI ordination axes. We used generalized least squares to simultaneously estimate phylogenetic signal (Pagel's Lambda) and other model parameters[79].

**Data availability.** The data set of this study is published by Willerslev et al.[12]. Additional data are available from the corresponding author upon request.

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

## Acknowledgements

We are grateful to F. de Bello for his constructive comments on the manuscript. The data production was supported by the European Union 6th framework project ECOCHANGE (GOCE-2006-036866, coordinated by P.T.). M.Z., J.D. and M.M. were supported by grants from the Estonian Research Council (IUT 20-28) and the European Regional Development Fund (Centre of Excellence EcolChange). C.B. was supported by the Research Council of Norway (grant no 191627/V40).

## Author contributions

M.M. and M.Z. designed the study. M.M., J.D. and C.B. compiled the trait database and J. D. performed the analyses of the data. M.Z., J.D., M.E. and M.M. wrote the first draft of the paper, C.B., P.T., E.W. and E.C. discussed the results and contributed to the writing of the final manuscript.

## Additional information

**Competing interests:** The authors declare no competing financial interests.

