## [Peer Review File · Nature Communications]

Reviewers' comments:

Reviewer #1 (Remarks to the Author):

The ms addresses a very interesting question, i.e. how interaction structures are assembled in past ecosystems. It does not of course look at entire networks, but only a small fraction, i.e. the (perhaps) mutualistic interactions between plants and their associated fungi. In addition, it makes a few broad inferences about pollination interactions. As such the ms is of high value. The results are unexpected, as were the results in a previous study about vegetation types in circumpolar regions (Villierslev et al. 2014). In order to accept the results one has, of course, to accept the data behind. Here, the authors use two sources, one about vegetation changes during the last 50 ky and one about host plant-fungi interactions extracted from literature and digital sources. Thus the answers presented cannot be better than those two sources. Below I list some of my main worries but also stress the ways the authors circumvent some of these methodological problems.

? Data sampling

Is it really known whether the sites from which plant-DNA was sampled (Villierslev et al. 2014) had permafrost during the 50 ky? If not this might introduce an error into your analysis, because of differential leaching of DNA through soil zones – an error also mentioned in Villierslev et al. 2014.

Vertical soil movements in polygon soil may mess up your samples. Although this point is addressed in Villierslev et al. 2014 by saying that this can be “accounted for by careful site selection and by excluding rare DNA sequence reads”. I do not know what Villierslev et al. 2014 means about “careful site selection” (what are the criteria?), and Villierslev et al. 2014 does not address the consequences of the last problem (rare sequences).

? Spatial heterogeneity of samples and conclusions

Villierslev et al. 2014 show that sample similarity decreases with distance. This heterogeneity should preclude data pooling, but in spite of that both the 2014 paper and the present one are mainly presenting circumpolar generalizations about the last 50 ky. The heterogeneity may obscure and bias the results (“The myth about the overall conformity of the arctic vegetation”). This absolutely critical issue is not addressed sufficiently in the present ms. Even locally, Villierslev et al. 2014 found that the vegetation was a mosaic of different habitats. This ms inherits these problems and uncertainties from Villierslev et al. 2014, and of course adds new ones.

? Pollination

The pollination part of the ms is in my opinion the weakest. As the ms also says: many arctic plants are known to have a mixed breeding system of insect-selfing or insect-wind and basically the conclusions are based on a simplified typological view of pollination, that most pollination biologists have left long ago.

? Descriptive study

The study is mainly descriptive and lacks any solid explanations for why we should see this change in the composition of fungi/plant diversity during the 50 Ky.

? Stable associations between fungi and plants & non-analogue systems

Using available published data about fungi etc. the authors assume that the associations are constant between host and associated organisms. This might seem odd given that the ms also talks about past non-analogue systems. Thus one would also expect non-analogue selection regimes acting upon the organisms of the systems.

Conclusions

My conclusions about the ms are mixed. It is an excellent idea to couple the information from Villierslev et al. 2014 and information about fungi and N-fixators. It gives us the next piece of information about the structure of ecosystems during the last 50 ky in the high north, e.g. in

relation to nutrient availability. However, I am not convinced about a sufficient data quality, especially because the authors have to refer to non-analogue systems, which to me just means incongruence between climate and biotic data (I am probably not fair, but to me it is too easy just to refer to previous “ghost” systems different from today – take that as a subjective comment). However, my main objections are about data sampling, spatial heterogeneity of results and generalizations being too sweeping.

Reviewer #3 (Remarks to the Author):

Zobel et al

The study reports on temporal changes of plant trait composition of arctic vegetation during the recent Quaternary (pre-LGM – LGM – post-LGM) based on previously published metagenomic data combined with recent plant trait data on mutualistic relationships collated from public databases. Uncertainty of trait information (most taxa were higher than species) was treated by appropriate randomizations. Authors find temporal changes of shares of growth form, mycorrhizal type, mycorrhizal status and pollination type. Most importantly, observed patterns did not confirm to predictions based on the relationships between current climatic clines and traits leading to the general conclusion that forecasting responses to climate change may be limited by our understanding of the effect of mutualistic relationships. Thus, overall the study uses up-to-date methods and is highly topical and principally suitable for the journal.

My main point is that of biased sample size (the following numbers are largely based on Tab. S2 of Willerslev et al. 2014 as no clear data on sample size is available for this papers data, which should be ameliorated). For the three time periods Pre-LGM/LGM/post-LGM there were:

numbers of samples: 150 / 32 / 61;

number of observations (MOTUs in samples): 700 / 147 / 191;

number of MOTUs: 108 / 44 / 73.

Thus, sampling effort differed strongly between the three periods (I actually wonder why this aspect was also not mentioned in the Willerslev paper). Although authors used the relative proportion of traits, highly unequal sample sizes may have affected the result, e.g. because of chance effects in the less well samples periods. In addition, it is known that the representation in eDNA differs between life forms which are correlated with the traits analysed here. Both effects could have influenced the contribution of MOTUs and traits in the three time steps. Consequently some kind of rarefaction analysis has to be performed, e.g. randomly picking for each time step 146 (min observations - 1) observations.

Overall I have the impression that authors favored a reasonable story over stringency, e.g. because non-significant changes of pollination mode are nevertheless discussed as they were.

Detailed comments

L21-22 “the representation of ectomycorrhizal, non-mycorrhizal, facultatively mycorrhizal and insect-pollinated taxa all increased” difficult to grasp the actual message because different classifications unknown to the reader are mixed (ecto and facultative...). Moreover at this point the three mutualisms to which the classifications refer to are not yet mentioned. Must be rewritten.

L202 “During the cool and dry LGM, plant species richness decreased” This generalisation is unjustified as it is based on biased, i.e. strongly unequal sample sizes. Rarefaction needs to be employed.

L206 “and that of insect-pollinated taxa increased” I do not understand: in L171 it was stated that there were no significant differences between time periods. Contradiction. (similarly L210/211)

Table 1. “MOTUs for which any traits were undefined were omitted from the data set.” Unclear because in the M+M section you write that all MOTUs with missing trait data were skipped from

the whole analysis. Clarify: how many traits had how many Nas in the final dataset?

Fig.2 I expected to see 131 points (e.g. 6 tree and 28 graminoids) in the OMI ordination; however very much less are visible (2 trees, 8 graminoids). If this is due to identical positions in the ordination space, symbol size should reflect data density. If this is due to other reasons, it must be explained.

Reviewer #4 (Remarks to the Author):

This paper builds upon a recent paper by Willerslev et al. (2014) in Nature, in which several million 'molecular operational taxon units' (MOTUs) from 242 Arctic aDNA samples were matched to modern plant species and used to estimate changes in plant diversity over the last 50,000 years. This manuscript extends Willerslev et al. by doing a trait-based analysis of the Willerslev et al. aDNA reads and MOTUs, with inferred plant species classified according to mycorrhizal strategy, pollination strategy, N-fixer vs. not, and life form.

Overall, I thought this was an interesting set of analyses, but did not find the findings and insights gained to be significant enough to be of wide interest to Nature Communication's readership. A key difficulty is that the 242 aDNA samples are grouped into three very broad time periods ('pre-LGM' – 50ka to 25ka, 'LGM' – 25 to 15 ka, and 'post-LGM' – 15ka to present) and have no spatial context at all.

This very coarse resolution makes it very hard to link the observed changes in trait frequencies to the known environmental and ecological changes in the Arctic over the last 50,000 years. There is a wealth of Arctic paleoclimatic and paleoecological records, with a detailed understanding of the spatial and temporal changes in climate, plant distributions and extinction, and timing of population crashes and extinction of megafauna. This information is unavoidably described in only the most general terms (LGM = cold and dry; post-LGM = warm and moist; pre-LGM = variable but somewhere in between).

Because the trait data are so spatiotemporally coarse and lack environmental context, it's hard to know whether the reported patterns are ecologically meaningful. For example, the paper reports that 'trait-environment relationships over geological time routinely deviated from those observed along contemporary gradients.' (L220-221) – I don't think this conclusion can be supported by the available data. Similarly, the interpretations come across as interesting but arm-waving. For example, the arguments about ruptured mycorrhizal partnerships (L247-258) and shifts from N- to P-limitation (L266-286) are interesting but highly speculative.

Second, the paper makes claims for changes in biotic interactions and mutualisms (e.g. lines 28-37), but the trait analyses can only provide indirect insight into these past changes. We only know, for example, that the relative frequencies of plant mycorrhizal hosts changed among the three time periods, and we have no independent information about distribution or diversity of the fungal symbionts. The title '...reveals highly dynamic mutualisms' comes across as an oversell.

A last key question is whether this paper represents enough of a new advance to warrant publication in Nature Communications. This data is founded on aDNA results that were published in Willerslev et al. (2014). The new contribution here is the trait analysis of the plant species reported by Willerslev et al. This is worthy, but I'm on the fence about whether it passes the bar for a Nature-family ms. Normally I would have said no, but my understanding is that Nature Communications are supposed to provide quick descriptions of smaller advances, so this may be appropriate. Up to the editors here.

Reviewers' comments:

Reviewer #1 (Remarks to the Author):

The ms addresses a very interesting question, i.e. how interaction structures are assembled in past ecosystems. It does not of course look at entire networks, but only a small fraction, i.e. the (perhaps) mutualistic interactions between plants and their associated fungi. In addition, it makes a few broad inferences about pollination interactions. As such the ms is of high value. The results are unexpected, as were the results in a previous study about vegetation types in circumpolar regions (Villierslev et al. 2014). In order to accept the results one has, of course, to accept the data behind. Here, the authors use two sources, one about vegetation changes during the last 50 ky and one about host plant-fungi interactions extracted from literature and digital sources. Thus the answers presented cannot be better than those two sources. Below I list some of my main worries but also stress the ways the authors circumvent some of these methodological problems.

R: *We appreciate the reviewer's positive view of the study and address the specific concerns below.*

? Data sampling

Is it really known whether the sites from which plant-DNA was sampled (Villierslev et al. 2014) had permafrost during the 50 ky? If not this might introduce an error into your analysis, because of differential leaching of DNA through soil zones – an error also mentioned in Villierslev et al. 2014.

R: *Virtually all the samples are derived from frozen yedoma deposits. Given that the Holocene is the warmest period in the record, and that the deposits are currently frozen, it is reasonable to assume that the source material has been continually frozen. Loess deposition leads to incremental accumulation of the deposit over time, with permafrost moving upwards in parallel. Villierslev et al (2014) acknowledge that any sample likely represents a period of time, rather than a (geological) instant. This period depends upon the depth of the active layer and the rate of accumulation; it represents time when the sampled material was in the active layer, and thus possibly subject to root penetration and cryoturbation, plus some degradation; it is also likely to represent decades to centuries of plant growth at the sampling site. As the data are divided into three broad temporal units, one can argue that this temporal integration within any one sample is a positive feature of the dataset.*

The large number of sediment-based studies of many kinds, including those from soils, which yield coherent records of molecules (e.g. isotopes, biomarkers) argue for downward leaching not being a problem.

Vertical soil movements in polygon soil may mess up your samples. Although this point is addressed in Villierslev et al. 2014 by saying that this can be “accounted for by careful site selection and by excluding rare DNA sequence reads”. I do not know what Villierslev et al. 2014 means about “careful site selection” (what are the criteria?), and Villierslev et al. 2014 does not address the consequences of the last problem (rare sequences).

R:

1. Careful site selection refers to ensuring the exposures of sediment were frozen when sampled (see above). Furthermore, the samples were taken well away from the contacts between ice wedges and sediment to ensure that material that may have been moved vertically in relation to frost-cracking and ice-wedge growth was not sampled.

2. The second point that Willerslev et al refer to - excluding rare DNA sequence reads - refers to the fact that MOTUs represented by very few reads have a relatively high probability of representing molecular artefacts generated during the laboratory preparation or sequencing of material and are thus omitted during bioinformatic treatment of the sequence data.

We now provide further detail in the manuscript on both of these points – L 321-351 (1) and L 358 (2).

? Spatial heterogeneity of samples and conclusions

Villierslev et al. 2014 show that sample similarity decreases with distance. This heterogeneity should preclude data pooling, but in spite of that both the 2014 paper and the present one are mainly presenting circumpolar generalizations about the last 50 ky. The heterogeneity may obscure and bias the results (“The myth about the overall conformity of the arctic vegetation”). This absolutely critical issue is not addressed sufficiently in the present ms.

Even locally, Villierslev et al. 2014 found that the vegetation was a mosaic of different habitats. This ms inherits these problems and uncertainties from Villierslev et al. 2014, and of course adds new ones.

R: We agree that this is a critical point and was an important omission from the earlier version. We see several related theoretical and technical issues connected with the broad area of spatial heterogeneity and how it might influence our conclusions: (i) whether sampling of heterogeneous vegetation mosaics renders global conclusions inappropriate; (ii) spatial autocorrelation, reflecting spatial dependence in (unmeasured) environmental variables and biogeographic processes – i.e. spatially proximate communities are more similar than spatially distant communities. This lack of independence can bias hypothesis tests because residual variance is underestimated; (iii) scale-dependence in measured trait-environment correlation – i.e. do proximate and distant communities exhibit the same trait responses to environmental change?

The samples indeed represent “snapshots” of the floristic composition of ancient vegetation, though each sample actually has greater temporal depth than spatial cover. We consider these snapshots valid for two reasons: first, the region studied (the unglaciated Arctic and subarctic of Eurasia and N America), while displaying floristic variation related to small-scale heterogeneity, was subject to similar regional climatic conditions during the three study periods, with the same key plant dominant functional types (e.g., all within the same biome). Furthermore, most samples come from yedoma, and thus reflect the upland (i.e. interfluvial) or zonal vegetation of the time, making the source material for samples more homogenous than if all landscape units (e.g., river valleys, mountaintops) were sampled evenly. We have added greater detail about this and other aspects of Willerslev et al.s sampling to the manuscript (L 339-351).

As the reviewer correctly points out, Willerslev et al. 2014 reported distance decay of similarity (taxonomic similarity of communities declined with distance), so autocorrelation might also be expected in this analysis of mutualist trait composition. To assess this we used Mantel correlograms. These indeed

indicated that taxonomic and trait composition was frequently positively autocorrelated at small distance classes and negatively at wide distance classes (Fig S1). However, the strength of autocorrelation was low (among all traits and distance classes, Mantel r always < 0.2). To investigate how this degree of autocorrelation might affect our hypothesis tests, we first used multi-scale ordination (MSO) to estimate to what extent residual variance was underestimated ($< 7\%$; numbers in Fig S1). We then calculated the theoretical relationship between variance underestimation and Type I error for our PERMANOVA analyses (Fig S2) and recalculated PERMANOVA with inflated variance estimates to the degree indicated by the MSO analysis (Table S1). These analyses strongly suggest that the effect of autocorrelation on our hypothesis tests is very low. We have added sections to the Methods (L 468-482), Results (L 147-152) and supplementary materials, to present these new analyses.

The reviewer is also correct that we presented general relationships that implied relevance throughout the Arctic study area. We agree that the question of scale-dependence in measured correlations is an important one to address explicitly and, where possible, to test. To do this we used the results from the MSO analysis to assess whether spatially-decomposed patterns of residual, explained and total variance were consistent with scale-independence. The close relationship between total variance and the sum of explained and residual variance at different spatial scales that we recorded for most traits is consistent with scale-independence; however, the patterns for growth form and mycorrhizal type were less consistent (Fig S3). We therefore draw an equivocal conclusion in the Discussion and add that the moderate level of data and its heterogeneous nature prevented us from investigating regional patterns in detail (L 215-221). For our own peace of mind we further reanalyzed the PERMANOVA analysis including an arbitrary regional distinction that best fitted the data set: North America; East Siberia [east of Longitude 120]; West Siberia [west of Longitude 120] (note that even this three region distinction reduced the sample size < 10 in several region \times climatic period groups). We found that the interaction term climatic period:region was never significant, though it was marginal $P < 0.1$ for a couple of traits. We do not present this analysis in the manuscript because the replication is low and the regional definition is debatable. However, it strengthens our impression that an equivocal conclusion is reasonable. We believe that our approach is analogous to presenting a main-effect model when replication does not permit an interaction term to be tested. The findings of the main effects model are not invalidated by the lack of an interaction term; but the main effect should be interpreted at the appropriate coarse scale.

? Pollination

The pollination part of the ms is in my opinion the weakest. As the ms also says: many arctic plants are known to have a mixed breeding system of insect-selfing or insect-wind and basically the conclusions are based on a simplified typological view of pollination, that most pollination biologists have left long ago.

R: *We completely agree that the pollination typology we use is simplified and does not provide detailed insight. However, it is the best we can do with the current data. Another option would be to exclude the pollination section from the manuscript altogether. We could do so if the reviewer or editor strongly feels that this is the correct course of action. At the same time, ours is the first report to describe representation of pollination types in ancient vegetation. Because many paleoecological studies are based on pollen analysis, which means that they rely largely on wind pollinated species, even a simplified description of the presence of different pollination types might be of use.*

? Descriptive study

The study is mainly descriptive and lacks any solid explanations for why we should see this change in the composition of fungi/plant diversity during the 50 Ky.

R: *We certainly agree that our study is descriptive. Hence we must be careful when presenting causal explanations as all such attempts inevitably carry the element of speculation. In the current manuscript, we offer several potential explanations for certain mutualist trait shifts in the Discussion. As we note above (in our reply to the Editor) we believe that making such points is reasonable, but that we have to be very clear that these are essentially hypotheses and are not being proposed with certainty. We have clarified the potential explanations we present in the Discussion and now explicitly present them as working hypotheses for future study.*

? Stable associations between fungi and plants & non-analogue systems

Using available published data about fungi etc. the authors assume that the associations are constant between host and associated organisms. This might seem odd given that the ms also talks about past non-analogue systems. Thus one would also expect non-analogue selection regimes acting upon the organisms of the systems.

R: *The mutualistic associations of plant species with mycorrhizal fungal taxa and N-fixing bacteria, are certainly not constant, but may change in space and time (Davison et al. 2015 Science 349:6251, Tedersoo et al 2014 Science 346:1078, Rodriguez-Echeverria 2010 Journal of Biogeography, 37:1611–1622.). Nonetheless, the basic nature of the relationship – whether a plant species is nonmycorrhizal, facultatively or obligately mycorrhizal, whether it’s mycorrhizal type is AM, ECM or ERM, or whether it associates with symbiotic N-fixing bacteria is a conservative trait that is likely to have evolved millions of years ago (Brundrett 2009 Plant and Soil 320:37-77; Werner et al. 2014 Nature Comm 5). The basic nature and type of symbiotic relationship is thus expected to remain the same under different environmental conditions. Importantly, there is one type of symbiotic relationship – Facultatively mycorrhizal status – where plant species are able either to facilitate or downregulate AM association depending on local environmental conditions (Smith & Read 2008, Mycorrhizal symbiosis, Moora 2014 J. Veg. Sci. 25, 1126-1132). We do not address this issue in the manuscript currently, but are ready to do so if the reviewers or Editor find it appropriate.*

4?) Conclusions

My conclusions about the ms are mixed. It is an excellent idea to couple the information from Villerslev et al. 2014 and information about fungi and N-fixators. It gives us the next piece of information about the structure of ecosystems during the last 50 ky in the high north, e.g. in relation to nutrient availability. However, I am not convinced about a sufficient data quality, especially because the authors have to refer to non-analogue systems, which to me just means incongruence between climate and biotic data (I am probably not fair, but to me it is too easy just to refer to previous “ghost” systems different from today – take that as a subjective comment). However, my main objections are about data sampling, spatial heterogeneity of results and generalizations being too sweeping.

R: We use the term "non-analogue" to refer to biomes/communities that do not have modern analogues, or at least spatially extensive ones. They are assumed to have occurred under abiotic conditions that have no modern analogues. A fundamental point is that this does not require incongruence between vegetation composition and climate; rather, the range and/or combination of past climatic and other abiotic conditions extends outside what we observe today, and for this reason we may not expect to accurately 'predict' past vegetation based on contemporary vegetation-environment relationships. Indeed palaeoecological research and modelling shows that over time plant functional types occur in the bioclimatic conditions that suit them. We have attempted to clarify what we mean by non-analogue and have added a more detailed description to the manuscript of how climate and other important environmental conditions are believed to have changed during the last 50000 years (L 208-210, 235-239, 246-248, 253-256, 339-351).

Reviewer #3 (Remarks to the Author):

Zobel et al

The study reports on temporal changes of plant trait composition of arctic vegetation during the recent Quaternary (pre-LGM – LGM – post-LGM) based on previously published metagenomic data combined with recent plant trait data on mutualistic relationships collated from public databases. Uncertainty of trait information (most taxa were higher than species) was treated by appropriate randomizations. Authors find temporal changes of shares of growth form, mycorrhizal type, mycorrhizal status and pollination type. Most importantly, observed patterns did not confirm to predictions based on the relationships between current climatic clines and traits leading to the general conclusion that forecasting responses to climate change may be limited by our understanding of the effect of mutualistic relationships.

Thus, overall the study uses up-to-date methods and is highly topical and principally suitable for the journal.

R: We appreciate the reviewer's positive overview.

My main point is that of biased sample size (the following numbers are largely based on Tab. S2 of Willerslev et al. 2014 as no clear data on sample size is available for this papers data, which should be ameliorated). For the three time periods Pre-LGM/LGM/post-LGM there were:

numbers of samples: 150 / 32 / 61;

number of observations (MOTUs in samples): 700 / 147 / 191;

number of MOTUs: 108 / 44 / 73.

Thus, sampling effort differed strongly between the three periods (I actually wonder why this aspect was also not mentioned in the Willerslev paper). Although authors used the relative proportion of traits, highly unequal sample sizes may have affected the result, e.g. because of chance effects in the less well samples periods. In addition, it is known that the representation in eDNA differs between life forms which are correlated with the traits analysed here. Both effects could have influenced the contribution of MOTUs and traits in the three time steps. Consequently some kind of rarefaction analysis has to be performed, e.g. randomly picking for each time step 146 (min observations - 1) observations.

Overall I have the impression that authors favored a reasonable story over stringency, e.g. because non-significant changes of pollination mode are nevertheless discussed as they were.

R: We used Willerslev et al.'s 2014 data set, which is certainly far from being ideally balanced and replicated, but it is still the most comprehensive data set available. That said, the reviewer is of course correct that the basis for robust analysis should first be established before drawing conclusions. On the question of sample sizes, we have added a supplementary analysis and figure showing how the multivariate precision of the composition estimates evolves with sample size (Fig S5). This shows that precision within each group improves most rapidly with increasing effort up to about 15 samples; by the time 30 samples are included, precision is relatively high (multivariate SE always < 0.2 compared with > 0.5 with 5 samples) and the rate of further increase is low. Consequently, all of the climatic period groups have relatively high and similar precision. We have also used bootstrapping to estimate 95% confidence intervals around the shares of trait categories in different climatic periods and we now show these in new Fig 1. This figure also indicates that error around estimates of compositional shares are reasonably low and similar between climatic periods. We therefore believe that observed group differences are unlikely to be biased by uneven sample sizes. For peace of mind we also recalculated PERMANOVA (read abundance) using 1000 randomly balanced datasets (i.e. within each time period, samples were randomly discarded until all periods were represented by 32 samples). Letter Figure 1 shows how the P values associated with these balanced analyses were distributed. The mean P value associated with the randomly balanced analysis was in each case very close to the observed P value calculated on the basis of unbalanced data. We do not plan to include this figure in the manuscript, since it already contains several sensitivity/alternative analyses, but could do so if the Editor or Reviewer feels strongly about this.

We revised the discussion of pollination mode and omitted discussing nonsignificant patterns. We made it clear that sometimes community level analysis (PERMANOVA) and species level ordination (OMI) gave slightly conflicting results (L 189-192; 291-298).

Letter Fig 1. P values derived from PERMANOVA analysis of mutualist trait composition (read abundance) in different climatic periods (pre-LGM, LGM, post-LGM) using randomly balanced data sets (N = 32 in all

periods; 1000 random iterations). Histogram bars show the frequency of P values in the balanced analyses, the red dashed lines show the mean P value among the balanced set; the black dashed lines show the observed P value in the corresponding analysis of unbalanced data (the black and red lines are essentially overlapping in the panels for growth form, mycorrhizal type and pollination).

Detailed comments

L21-22 “the representation of ectomycorrhizal, non-mycorrhizal, facultatively mycorrhizal and insect-pollinated taxa all increased” difficult to grasp the actual message because different classifications unknown to the reader are mixed (ecto and facultative...). Moreover at this point the three mutualisms to which the classifications refer to are not yet mentioned. Must be rewritten.

R: *We agree and apologise for the lack of clarity. We have rewritten this paragraph.*

L202 “During the cool and dry LGM, plant species richness decreased” This generalisation is unjustified as it is based on biased, i.e. strongly unequal sample sizes. Rarefaction needs to be employed.

R: *We apologise for not clearly presenting the evidence to support this statement. We now refer to Willerslev et al. 2014 and the jackknife estimates of richness that they generated to account for differences in sampling effort (L 182).*

L206 “and that of insect-pollinated taxa increased” I do not understand: in L171 it was stated that there were no significant differences between time periods. Contradiction. (similarly L210/211)

R: *Thank you for pointing this out! We removed the contradictions and rewrote those sections.*

Table 1. “MOTUs for which any traits were undefined were omitted from the data set.” Unclear because in the M+M section you write that all MOTUs with missing trait data were skipped from the whole analysis. Clarify: how many traits had how many Nas in the final dataset?

R: *We agree that this information was not clear and we did our best to clarify this issue in the manuscript. After compiling the trait database, we removed 14 MOTUs (2226214 reads) that could not be defined for all traits (five traits) and five samples that no longer contained any MOTU data. This left a data table containing 216 samples (145 pre-LGM, 32 LGM, 39 post-LGM), 131 MOTUs and 5026792 reads (L 397-401).*

Fig.2 I expected to see 131 points (e.g. 6 tree and 28 graminoids) in the OMI ordination; however very much less are visible (2 trees, 8 graminoids). If this is due to identical positions in the ordination space, symbol size should reflect data density. If this is due to other reasons, it must be explained.

R: *The reviewer is correct that the apparent lack of points is due to identical positioning in ordination space. We have recreated this figure scaling symbol size to reflect data density, as suggested (Fig 2).*

Reviewer #4 (Remarks to the Author):

This paper builds upon a recent paper by Willerslev et al. (2014) in Nature, in which several million 'molecular operational taxon units' (MOTUs) from 242 Arctic aDNA samples were matched to modern plant species and used to estimate changes in plant diversity over the last 50,000 years. This manuscript extends Willerslev et al. by doing a trait-based analysis of the Willerslev et al. aDNA reads and MOTUs, with inferred plant species classified according to mycorrhizal strategy, pollination strategy, N-fixer vs. not, and life form.

Overall, I thought this was an interesting set of analyses, but did not find the findings and insights gained to be significant enough to be of wide interest to Nature Communication's readership. A key difficulty is that the 242 aDNA samples are grouped into three very broad time periods ('pre-LGM' – 50ka to 25ka, 'LGM' – 25 to 15 ka, and 'post-LGM' – 15ka to present) and have no spatial context at all.

R: We appreciate the reviewer's concerns. We were indeed limited by the existing data set. It does not have ideal coverage and depth, but it is by a long distance the most complete (relatively unbiased) and detailed data source available to us. Concerning the spatial context, we performed further analyses to address spatial structure in the data (see the response to Reviewer 1 above). On the basis of these we are satisfied that our hypothesis tests are robust. However, they are undoubtedly coarse, and despite identifying no evidence of spatial dependency in the trait-climate relationships we cannot rule out the existence of regional differences due to the relatively sparse data set. Please also see our response above regarding the coherence of the sampling region in terms of its palaeobiogeography and the nature of the samples (most derived from upland, i.e., interfluvial positions).

This very coarse resolution makes it very hard to link the observed changes in trait frequencies to the known environmental and ecological changes in the Arctic over the last 50,000 years. There is a wealth of Arctic paleoclimatic and paleoecological records, with a detailed understanding of the spatial and temporal changes in climate, plant distributions and extinction, and timing of population crashes and extinction of megafauna. This information is unavoidably described in only the most general terms (LGM = cold and dry; post-LGM = warm and moist; pre-LGM = variable but somewhere in between).

R: We agree that the density of data points does not allow us to address relationships between ancient plant community trait spectra and known environmental parameters with the precision that we would like. However, as far as we are aware, it is currently the only existing plant community composition data set encompassing the last 50,000 years (as compared with, for example, pollen databases that typically address the past 21,000 years at most). We have done our best to exploit this dataset for what it can deliver, which is a broad comparison of three different phases of the late Quaternary (part of MIS 3, MIS 2 and MIS 1), while avoiding 'sweeping generalizations' that are not supported by the data.

Because the trait data are so spatiotemporally coarse and lack environmental context, it's hard to know whether the reported patterns are ecologically meaningful. For example, the paper reports that 'trait-environment relationships over geological time routinely deviated from those observed along contemporary gradients.' (L220-221) – I don't think this conclusion can be supported by the available data. Similarly, the interpretations come across as interesting but arm-waving. For example, the arguments about ruptured mycorrhizal partnerships (L247-258) and shifts from N- to P-limitation (L266-286) are interesting but highly speculative.

R: *We agree that the data we have do not allow us to draw far reaching conclusions, even about correlative relationships, let alone causal relationships. At the same time, as mentioned by Reviewer 1, there is a need to offer some potential explanations for the observed patterns. We have toned down the interpretation of our results and state clearly that suggestions concerning the possible ecological mechanisms underlying observed patterns are hypotheses, rather than established mechanistic explanations (L 212-214).*

Second, the paper makes claims for changes in biotic interactions and mutualisms (e.g. lines 28-37), but the trait analyses can only provide indirect insight into these past changes. We only know, for example, that the relative frequencies of plant mycorrhizal hosts changed among the three time periods, and we have no independent information about distribution or diversity of the fungal symbionts. The title ‘...reveals highly dynamic mutualisms’ comes across as an oversell.

R: *We also agree that we currently lack any information about ancient fungal communities, communities of N-fixing bacteria and pollinators. We anticipate that this information will become available in the future, and agree that this would allow fascinating and comprehensive analysis. However, we believe that a robust understanding of the nature and biogeography of mutualistic relationships can be obtained based on the study of a single partner and note that this approach has an established history (e.g. Read, D. J. & Perez-Moreno, J. Mycorrhizas and nutrient cycling in ecosystems - a journey towards relevance? *New Phytol.* 157, 475-492 (2003), Bueno et al. *Global Ecol Biog* 2017 DOI: 10.1111/geb.12582).*

A last key question is whether this paper represents enough of a new advance to warrant publication in Nature Communications. This data is founded on aDNA results that were published in Willerslev et al. (2014). The new contribution here is the trait analysis of the plant species reported by Willerslev et al. This is worthy, but I’m on the fence about whether it passes the bar for a Nature-family ms. Normally I would have said no, but my understanding is that Nature Communications are supposed to provide quick descriptions of smaller advances, so this may be appropriate. Up to the editors here.

R: *We appreciate this comment. We think that such an approach – combining aDNA data with plant trait information - is novel and may serve as a template for future studies.*

Reviewers' comments:

Reviewer #3 (Remarks to the Author) (Same reviewer as the original round of review):

Zobel et al revision

In the previous version I had criticized the somewhat generous treatment of significance levels and the lack of rarefaction analyses given the strongly different sample sizes in the three time periods.

Both issues have been addressed satisfactorily by more careful wording and by adding a rarefaction analysis as supplementary material.

Thus, from my side there are no further obstacles preventing acceptance.

Reviewer #5 (Remarks to the Author) (replacement reviewer for reviewer #4):

This paper leverages an existing dataset on plant diversity across the last 50,000 years of global change in two unglaciated Arctic regions to test whether changes in plant communities were also associated with changes in functional mutualist relationships.

Biotic interactions in paleorecords is topic of great interest, and this study adds an interesting angle by looking at pollinators and fungal relationships. As a paleoecologist with expertise in Quaternary vegetation dynamics, megafaunal extinctions, functional ecology, and biotic interactions, I have focused my comments accordingly.

Overall, I think this paper shows promise. It's exciting to see this amazing dataset leveraged in a new and exciting way, and this is certainly a question of great interest, particularly in the era of global change. However, there are some issues that weaken this paper and undermine its message. I have some general comments first, followed by some line-specific edits.

First, the paper is set up/justified by citing the stability of mutualisms as a central question, yet does not necessarily address that question with these data. Relationships with mutualists are inferred from contemporary patterns. That's not necessarily a fatal flaw in and of itself. However, the authors refer to mycorrhizal associations as mutualisms, when this was likely not the case during the majority of the period of interest. During full glacial conditions represented by this study, CO₂ concentrations were as low as 180 ppm; at this point, many plants are nearing carbon starvation, and mycorrhizal associations were likely parasitic, and not mutualistic. Indeed, CO₂ is only given a cursory mention in this paper, when its relationship with climate and other global change factors would have been significant, including in terms of impacting growth forms and competition. This makes the interpretations much more tentative, and I read most of the paper feeling like CO₂ was the mammoth in the room, so to speak.

A second major issue I had was that the paper lacks key references and does not do a sufficient job of positioning itself relative to the broader literature. There have been a range of recent studies outlining biotic interactions in the Quaternary; this paper does not cite them. I've highlighted several places where a better grounding in the literature would improve this paper. Overall, the paper refers vaguely to drivers of the observed changes, but these aren't always clear. A stronger grounding in the literature will improve discussion of the observed patterns.

The inclusion of plant-pollinator relationships is tenuous and feels tacked on. I see that this was mentioned in an earlier round of reviews; I'm not convinced that prior concerns were fully addressed here. Indeed, the pollinator study doesn't seem to be as strong or compelling as the mycorrhizal analysis, so it might be worth jettisoning it (or beefing it up in the manuscript).

Please clarify the spatial, temporal, and taxonomic grain of this analysis early – was it continuous

throughout the last 50,000 years? All of the Arctic or just a few point locations from within a region? Only plant data (e.g., were other types of data inferred)? This is unclear even by the time I started reading the results, and needs to be established early on. Don't assume I've read the earlier Willerslev paper.

The discussion would benefit from some subheadings and better sign-posting overall.

The conclusion isn't necessarily set up well in the preceding text. Also, mycorrhizal associations are treated as a functional trait, which assumes that these relationships were conservative through the last 50,000 years. We don't actually know much about functional structure overall from this study, so be careful not to oversell this. The conclusion that "temperature changes were probably secondary to other drivers of change" is not necessarily supported from the text, as there is no independent paleoclimate proxy for this analysis, and this was not (from what I could tell) quantitatively determined. As written, the Conclusion is vague (e.g., "in other cases, changes were more unexpected..."). This section needs to be strengthened, better built up by the preceding paper, and clearer.

29 Cite Blois et al 2013 review in Science (or similar) here.

35 This needs a different reference for a non-specialist audience; Birks and Birks is methodological paper, and is not a review of paleovegetation data per se.

36 I suggest citing Neotoma (EPD is one of the databases in Neotoma, which also includes other global databases for pollen and plant macrofossils).

37 Revise to "However, such records are biased towards wind-pollinated taxa and high pollen producers, which can obscure..." Also, low pollen counts aren't the problem per se; pollen counts are relative, are determined quantitatively based on your research question and the study system, as well as rarefaction.

41 Another issue is that plant macrofossils typically record plants growing directly around core sites, while pollen typically represents vegetation at the watershed scale.

46 eDNA may be highly abundant in some sediments, but is not well-preserved in all climates and for much of the Quaternary. Each proxy has its strengths and weaknesses, and they complement one another in our understanding of the earth system; this introduction seems to suggest (incorrectly) that paleovegetation proxy data are weak and eDNA in sediments is the solution. Maybe in the Arctic, where preservation in permafrost is good – but even then, freeze-thaw action can be a problem. It's not necessary to oversell eDNA at the expense of pollen to make this paper novel.

56 This would be a good opportunity to cite the several papers that have come out in recent years on megafaunal-vegetation impacts (Bakker et al 2016, Gill et al 2009, Johnson et al 2009, Mahli et al 2015, etc.).

57 This paper is set up as a response/continuation of Willerslev et al, but I think it would be stronger to link it with the broader literature on biotic interactions in the Quaternary (including the citations above) and then highlight the fact that, while megafauna broadly have received a lot of attention, other biotic interactions have not. I'd also cite Wilkinson et al 1998 (GEB), Chapin et al 1996 (JVS) and others as evidence as to why this is important.

72 See Taylor et al 1995, Mycologia, and Stubblefield et al, 1987, Science.

75 See also the Wilkinson et al 1998 GEB paper mentioned previously.

86 See Gulbranson et al 2017, Geology

88 This transition feels very abrupt, and was not integrated into the paper well.

99 This assumes that these associations were stable through time. How well-justified is this in the contemporary global change literature?

100 This is unclear – are you referring to species' abundances? Species-climate relationships? Some community or biodiversity metric of interest?

102 "Shares" is unclear. Relative abundance?

111 MOTU has been used without being defined first. As this paper will be read by a broad audience unfamiliar with molecular methods, it would be useful to identify this first.

123 "Northern" needs some anchoring here. Do you mean Arctic broadly?

147 Thank you for including this section.

154 What is the benefit of OMI versus other ordination methods? Are axes interpreted quantitatively or qualitatively? I see that this is covered in the methods but a little extra info here to help with interpretation would be useful.

178 This is unsurprising given the magnitude of global changes in this region over this time—a brief context or summary of known vegetation shifts would benefit non-Quaternarist readers, either in the introduction (by way of setting up your hypotheses) or here. Ideally both.

182 What are your climate data? "Cool and dry" relative to...today? Are permafrost environments every really "dry?" This needs a little unpacking and some references.

192 This makes sense in the context of the loss of disturbance-associated forbs, allowing for grass and more woody taxa expansion following megafaunal extinction (Bakker et al 2016).

197 Here is where thinking about CO₂ would be really helpful – contemporary climatic gradients provide an climate-only null model, while a major difference would have been CO₂ concentrations.

201 A-ha! CO₂ finally gets mentioned here – this needs to come out earlier.

201 The records in (29) were largely temperate and Holocene.

208 See also all the ecosystem functions provided by Pleistocene megafauna...

209 (29) is not the appropriate reference here. See the CCSM3 transient paleoclimate simulations (Liu et al 2009 Science and follow-ups).

225 But there will also be climate-mediated changes in competitive interactions as well, and species-climate relationships are likely to change – particularly in no-analog environments such as are represented in the Arctic during your study interval.

236 There are regional and global proxy and modeled paleoclimate simulations that would better contextualize these interpretations – it would be best to cite them here.

240 This was not the case everywhere. Also, please make dates in chronologic order (e.g., 14 to 12 ka BP is the convention – ka (kiloannum) is better than ky, which isn't an official temporal unit). I also recommend all date windows be indicate with "to" rather than "-" which is also a

subtraction sign. It helps keep things clear.

250 Any information on Arctic grasslands? Those would be more appropriate to cite here.

258 Bogs are pretty unique systems, and are particularly nutrient poor, so I'm not sure if this is the best reference here.

264 "Post-LGM" is just "the Holocene."

266 Jack Williams' no-analog vegetation papers (2001, 2004, 2007) might be more appropriate for North America here.

267 Do you mean Beringia here specifically?

269 Again, cite Williams – these are well-established from pollen data as no-analog communities.

270 "allied" is a weird word choice here. Associated?

270 This needs some unpacking – your definition of LGM is a bit broad. Do you mean abrupt climate changes during deglaciation specifically? Disrupted food webs following megafaunal extinctions? What do you mean by stochastic events, and why would these have a greater impact at this time than any other? Also, do you mean dispersal lags, rather than bottlenecks per se?

271 We know from the Quaternary paleoecological record that species exhibited a highly individualistic response to changing climates throughout the late Quaternary, and that species associations (at least for plants and mammals) were not stable. Is it reasonable to expect species to respond in "highly coevolved" ways?

273 A competitive disadvantage given changing environmental conditions? Can you frame this in terms of the known environmental changes at this site?

279 This paper concluded (especially in the associated press) that changes in vegetation were a cause of megafaunal extinctions, rather than a consequence. There are a number of other papers that should be cited here: Bakker et al 2016, Gill et al 2009, Mahli et al 2016, Gill et al 2013, Johnson et al 2009, Barnosky et al 2016, Doughty et al 2016, and others. This statement is agnostic in terms of cause-and-effect, but this paper needs to be centered within the broader literature on megafauna impacts on vegetation, which is missing from this paper.

293 I'm not entirely convinced this isn't an artifact of binning – the climatic periods in this analysis include periods of climate variability in some cases.

321 Beringia?

327 Please include these ages in a supplemental table.

332 MIS 2 is 29 to 15 ka BP, and isn't necessarily synonymous with "LGM," which differs by region and represents a specific point at which glaciers reached their maximum extent. Similarly, "pre-LGM" is not synonymous with MIS 3, and your window (~60 to 25 ka BP) includes late MIS 4 and MIS 3. I'm not trying to nitpick here so much as to say that these time bins need to be clearer if they are to be meaningful. If you're using the terminology of defined stratigraphic markers, it's important to be precise.

Reviewer #6 (Remarks to the Author) (replacement for reviewer #1):

[Note from the editor: Reviewer #6 confidentially concluded that, overall, the responses to comments by reviewer #1 were satisfactory.]

My major comment is that there is a third alternative interpretation of the main findings, in addition to disappearing of megaherbivores, and "potential rupturing of mutualism" (which is a quite obscure mechanism, and could benefit from some explanation). It is now well-known that plant species respond individualistically to climate change, and that what we see as integrated plant communities is merely a mental construction by us, relatively short-lived humans. Thus, it may seem as plant communities during the late Pleistocene have been 'disrupted', while in fact everything represents a continuous process of change affecting individual species (due to climate, interactions with other organisms, pure chance events etc). The pattern detected in this study may represent a similar phenomenon. Different plant species (with different mycorrhiza) respond to the same factors as today, but also to other still unknown factors, and over the time period covered (which has been sliced into three distinct periods) this appear as a 'rupturing' of the mycorrhizal communities.

Minor comments

Since pollination is included in the results and discussion, it should also be mentioned in the Abstract.

I read the Abstract as a suggestion that the disappearance of megaherbivores may be a cause behind the changing community-level distribution of mycorrhizal types. In Willerslev et al. (2014) one gets the impression that the authors suggest it to be the other way around: the megaherbivores may have declined due to changing plant communities. In the text the authors make a remark on what (I interpret) they actually mean: that there is a positive feedback between herbaceous communities and megaherbivores, i.e. no simple cause and effect. This could be more clearly expressed in the Abstract.

On line 160, it is stated: "non-confounding mutualist traits". Can you please clarify what you mean by this.

Reviewers' comments:

Reviewer #3 (Remarks to the Author) (Same reviewer as the original round of review):

Zobel et al revision

In the previous version I had criticized the somewhat generous treatment of significance levels and the lack of rarefaction analyses given the strongly different sample sizes in the three time periods.

Both issues have been addressed satisfactorily by more careful wording and by adding a rarefaction analysis as supplementary material.

Thus, from my side there are no further obstacles preventing acceptance.

Thank you very much indeed!

Reviewer #5 (Remarks to the Author) (replacement reviewer for reviewer #4):

This paper leverages an existing dataset on plant diversity across the last 50,000 years of global change in two unglaciated Arctic regions to test whether changes in plant communities were also associated with changes in functional mutualist relationships. Biotic interactions in paleorecords is topic of great interest, and this study adds an interesting angle by looking at pollinators and fungal relationships. As a paleoecologist with expertise in Quaternary vegetation dynamics, megafaunal extinctions, functional ecology, and biotic interactions, I have focused my comments accordingly. Overall, I think this paper shows promise. It's exciting to see this amazing dataset leveraged in a new and exciting way, and this is certainly a question of great interest, particularly in the era of global change.

We are very grateful for the positive words!

However, there are some issues that weaken this paper and undermine its message. I have some general comments first, followed by some line-specific edits.

First, the paper is set up/justified by citing the stability of mutualisms as a central question, yet does not necessarily address that question with these data. Relationships with mutualists are inferred from contemporary patterns. That's not necessarily a fatal flaw in and of itself. However, the authors refer to mycorrhizal associations as mutualisms, when this was likely not the case during the majority of the period of interest. During full glacial conditions represented by this study, CO₂ concentrations were as low as 180 ppm; at this point, many plants are nearing carbon starvation, and mycorrhizal associations were likely parasitic, and not mutualistic. Indeed, CO₂ is only given a cursory mention in this paper, when its relationship with climate and other global change factors would have been significant, including in terms of impacting growth forms and competition. This makes the interpretations much more tentative, and I read most of the paper feeling like CO₂ was the mammoth in the room, so to speak.

We completely agree with this point and now note the possible role of CO₂ in driving changes in mycorrhizal type and status and the N fixing trait (abstract, L 103-105, L 111, L 144, L 241-245, L 264). We also agree that theory would suggest that under conditions of carbon starvation, the

mutualistic mycorrhizal relationship may become commensalistic or parasitic, although we are unaware of experimental evidence demonstrating this. We have now noted this in the text (L 103-105).

A second major issue I had was that the paper lacks key references and does not do a sufficient job of positioning itself relative to the broader literature. There have been a range of recent studies outlining biotic interactions in the Quaternary; this paper does not cite them. I've highlighted several places where a better grounding in the literature would improve this paper. Overall, the paper refers vaguely to drivers of the observed changes, but these aren't always clear. A stronger grounding in the literature will improve discussion of the observed patterns.

After the first round of reviewing, we reduced the number of references from 75 to 52 in order to follow the nominal journal guidelines. However, we certainly agree that adding some further references could improve the general framework of the paper. The new references recommended by this referee are very appropriate and we have tried to integrate many of them (though not all – due to the constraints set by the journal).

The inclusion of plant-pollinator relationships is tenuous and feels tacked on. I see that this was mentioned in an earlier round of reviews; I'm not convinced that prior concerns were fully addressed here. Indeed, the pollinator study doesn't seem to be as strong or compelling as the mycorrhizal analysis, so it might be worth jettisoning it (or beefing it up in the manuscript).

We agree that the analysis of pollination types in vegetation is the least satisfactory aspect, in terms of the categorization possible and lack of a marked trend in the data. We do not see much possibility for 'beefing it up', without excessive speculation. For these reasons, we have ourselves repeatedly considered omitting the analysis. The only reason we have not done so is that we believe the results, limited as they are, might nonetheless be of interest to palaeoecologists. Most analysis of historical vegetation is based on pollen, and for this reason, understanding how the relative abundance of wind pollinators (the predominating pollen producers) has changed in time is important (for example, it can inform about the coverage achieved by pollen analysis from different time periods). Because of this we suggest retaining the analysis, but we would accept removing this part if the editor or referee feel strongly that it should be removed.

Please clarify the spatial, temporal, and taxonomic grain of this analysis early – was it continuous throughout the last 50,000 years? All of the Arctic or just a few point locations from within a region? Only plant data (e.g., were other types of data inferred)? This is unclear even by the time I started reading the results, and needs to be established early on. Don't assume I've read the earlier Willerslev paper.

Here and later we have tried to avoid too much repetition of Willerslev et al's 2014 results, which are based on exactly on the same empirical material and published in the sister journal 'Nature'. However, we agree that some repetition is necessary so we have added more information of the sort requested by the reviewer in order to adequately describe the framework of the study (L 54-78, L 347-385).

The discussion would benefit from some subheadings and better sign-posting overall.

We agree with this, but unfortunately the format of Nature Communications does not allow subheadings. We have nonetheless worked to improve the flow of the Discussion (and Introduction).

The conclusion isn't necessarily set up well in the preceding text. Also, mycorrhizal associations are treated as a functional trait, which assumes that these relationships were conservative through the last 50,000 years. We don't actually know much about functional structure overall from this study, so be careful not to oversell this. The conclusion that "temperature changes were probably secondary to other drivers of change" is not necessarily supported from the text, as there is no independent paleoclimate proxy for this analysis, and this was not (from what I could tell) quantitatively determined. As written, the Conclusion is vague (e.g., "in other cases, changes were more unexpected..."). This section needs to be strengthened, better built up by the preceding paper, and clearer.

We agree and have rewritten the Conclusions, along with other parts of the Discussion and Introduction sections.

29 Cite Blois et al 2013 review in Science (or similar) here.

Thank you – we have included the citation.

35 This needs a different reference for a non-specialist audience; Birks and Birks is methodological paper, and is not a review of paleovegetation data per se.

Thank you for noticing this. We have replaced the Birks and Birks reference with Bradshaw, 2013; Birks, 2013; Grimm et al., 2013.

36 I suggest citing Neotoma (EPD is one of the databases in Neotoma, which also includes other global databases for pollen and plant macrofossils).

Thank you, we replaced the citation.

37 Revise to "However, such records are biased towards wind-pollinated taxa and high pollen producers, which can obscure..." Also, low pollen counts aren't the problem per se; pollen counts are relative, are determined quantitatively based on your research question and the study system, as well as rarefaction.

The revision has been made.

41 Another issue is that plant macrofossils typically record plants growing directly around core sites, while pollen typically represents vegetation at the watershed scale.

We agree and have made the revision.

46 eDNA may be highly abundant in some sediments, but is not well-preserved in all climates and for much of the Quaternary. Each proxy has its strengths and weaknesses, and they complement one another in our understanding of the earth system; this introduction seems to suggest (incorrectly) that paleovegetation proxy data are weak and eDNA in sediments is the solution. Maybe in the Arctic, where preservation in permafrost is good – but even then, freeze-thaw action can be a problem. It's not necessary to oversell eDNA at the expense of pollen to make this paper novel.

We agree. We have revised the text to describe the strengths and weaknesses of the different approaches more clearly and to specify more precisely the benefits of the eDNA approach given the geographical and temporal context of the study (L 54-61)

56 This would be a good opportunity to cite the several papers that have come out in recent years on megafaunal-vegetation impacts (Bakker et al 2016, Gill et al 2009, Johnson et al 2009, Mahli et al 2015, etc.).

Thank you, these are very relevant references. We have included some of them but unfortunately not all because we have already considerably exceeded the nominal limit of references.

57 This paper is set up as a response/continuation of Willerslev et al, but I think it would be stronger to link it with the broader literature on biotic interactions in the Quaternary (including the citations above) and then highlight the fact that, while megafauna broadly have received a lot of attention, other biotic interactions have not. I'd also cite Wilkinson et al 1998 (GEB), Chapin et al 1996 (JVS) and others as evidence as to why this is important.

Again, these are very good references indeed, but due to space constraints we have included only one of them (Wilkinson).

72 See Taylor et al 1995, Mycologia, and Stubblefield et al, 1987, Science.

We apologise for a lack of clarity. We meant the late-Quaternary but not earlier geological periods. We have clarified this in the text (L 94).

75 See also the Wilkinson et al 1998 GEB paper mentioned previously.

Thank you. We have included the citation.

86 See Gulbranson et al 2017, Geology

Thank you. We have included the citation.

88 This transition feels very abrupt, and was not integrated into the paper well.

We have rewritten this section.

99 This assumes that these associations were stable through time. How well-justified is this in the contemporary global change literature?

As mentioned above, we are unaware of empirical evidence about how the mutualistic interactions between plants and fungi change under carbon starvation. But it is quite probable that they became either commensalistic (+0) or even parasitic (+-). We now mention this possibility in the text (L 102-105). There is more information about the effects of high CO₂ on mycorrhiza – and some new references to such studies have been included (L 103).

100 This is unclear – are you referring to species' abundances? Species-climate relationships? Some community or biodiversity metric of interest?

We have changed the wording of this paragraph and the questionable section is no longer included.

102 “Shares” is unclear. Relative abundance?

We now use ‘relative abundance’.

111 MOTU has been used without being defined first. As this paper will be read by a broad audience unfamiliar with molecular methods, it would be useful to identify this first.

We added the definition (L 141).

123 “Northern” needs some anchoring here. Do you mean Arctic broadly?

We now describe the study area more clearly (L 55-56) and use more precise terminology throughout: such as ‘northern high latitudes’ (and not ‘Arctic’, as this does not incorporate all the studied areas).

147 Thank you for including this section.

Thank you very much for the positive comment!

154 What is the benefit of OMI versus other ordination methods? Are axes interpreted quantitatively or qualitatively? I see that this is covered in the methods but a little extra info here to help with interpretation would be useful.

We have added a little more explanation to the results (L 182-185).

178 This is unsurprising given the magnitude of global changes in this region over this time—a brief context or summary of known vegetation shifts would benefit non-Quaternarist readers, either in the introduction (by way of setting up your hypotheses) or here. Ideally both.

We added more information in the Introduction describing global changes and vegetation shifts (L 64-77) and changed the wording at the start of the Discussion to reiterate the magnitude of changes that occurred (L 225-228).

182 What are your climate data? “Cool and dry” relative to...today? Are permafrost environments every really “dry?” This needs a little unpacking and some references.

We elaborated the wording of this section (L 225-228) and also provide a referenced description of conditions in the Introduction (L 64-77).

192 This makes sense in the context of the loss of disturbance-associated forbs, allowing for grass and more woody taxa expansion following megafaunal extinction (Bakker et al 2016).

We apologize for the lack of clarity. We meant inconsistency in the statistical support, which may have been due to the small sample. We have changed the wording and now only note the sample size later in the text (L 315-316).

197 Here is where thinking about CO₂ would be really helpful – contemporary climatic gradients provide an climate-only null model, while a major difference would have been CO₂ concentrations.

We agree and have revised the text.

201 A-ha! CO₂ finally gets mentioned here – this needs to come out earlier.

We agree and have introduced CO₂ much earlier, in the Introduction (L 103-105).

201 The records in (29) were largely temperate and Holocene.

We agree and have revised the text.

208 See also all the ecosystem functions provided by Pleistocene megafauna...

We agree and have revised the text and added references.

209 (29) is not the appropriate reference here. See the CCSM3 transient paleoclimate simulations (Liu et al 2009 Science and follow-ups).

We have replaced the references.

225 But there will also be climate-mediated changes in competitive interactions as well, and species-climate relationships are likely to change – particularly in no-analog environments such as are represented in the Arctic during your study interval.

We apologise but we do not entirely follow the suggestion here. We agree with the points the reviewer raises, but do not see how they they should be incorporated into the text

236 There are regional and global proxy and modeled paleoclimate simulations that would better contextualize these interpretations – it would be best to cite them here.

We agree and have revised the text and added a reference (L225-228).

240 This was not the case everywhere. Also, please make dates in chronologic order (e.g., 14 to 12 ka BP is the convention – ka (kiloannum) is better than ky, which isn't an official temporal unit). I also recommend all date windows be indicate with "to" rather than "-" which is also a subtraction sign. It helps keep things clear.

We agree and have changed the text accordingly.

250 Any information on Arctic grasslands? Those would be more appropriate to cite here.

Unfortunately, there is no information about Arctic grasslands. In order to avoid confusion, we removed this section.

258 Bogs are pretty unique systems, and are particularly nutrient poor, so I'm not sure if this is the best reference here.

The term 'bog' was used by the authors of this paper but arguably not in a very precise way. Having worked in the same area ourselves, we would rather describe it as subarctic tundra scrub on peaty soil. Hence we would like to keep the reference, but omit the term 'bog' from the text.

264 "Post-LGM" is just "the Holocene."

We're not completely sure we've understood this comment. But since post-LGM and Holocene are not entirely equivalent (the latter starting several thousand years later), we would prefer to keep the original terminology.

266 Jack Williams' no-analog vegetation papers (2001, 2004, 2007) might be more appropriate for North America here.

We agree and have added a reference to the 2007 work but not to others, just due to the space constraints.

267 Do you mean Beringia here specifically?

We actually meant to refer to the sites studied in this manuscript and Willerslev et al, which cover a larger area than Beringia. We have clarified this in the text.

269 Again, cite Williams – these are well-established from pollen data as no-analog communities.

Done.

270 “allied” is a weird word choice here. Associated?

We have replaced ‘allied’ with ‘associated’.

270 This needs some unpacking – your definition of LGM is a bit broad. Do you mean abrupt climate changes during deglaciation specifically? Disrupted food webs following megafaunal extinctions? What do you mean by stochastic events, and why would these have a greater impact at this time than any other? Also, do you mean dispersal lags, rather than bottlenecks per se?

Thank you. We have thoroughly rewritten the text (L 281-294) and hope our explanation is now clearer.

271 We know from the Quaternary paleoecological record that species exhibited a highly individualistic response to changing climates throughout the late Quaternary, and that species associations (at least for plants and mammals) were not stable. Is it reasonable to expect species to respond in “highly coevolved” ways?

We have rewritten this section, maintaining the concept of preferred mutualist partners, but removing references to highly-coevolved organisms, which we agree is questionable speculation.

273 A competitive disadvantage given changing environmental conditions? Can you frame this in terms of the known environmental changes at this site?

The intended sense was that organisms might be at a competitive disadvantage if they were lacking mutualist partners. We rewrote this section and for clarity omitted the reference to competitive disadvantage.

279 This paper concluded (especially in the associated press) that changes in vegetation were a cause of megafaunal extinctions, rather than a consequence. There are a number of other papers that should be cited here: Bakker et al 2016, Gill et al 2009, Mahli et al 2016, Gill et al 2013, Johnson et al 2009, Barnosky et al 2016, Doughty et al 2016, and others. This statement is agnostic in terms of

cause-and-effect, but this paper needs to be centered within the broader literature on megafauna impacts on vegetation, which is missing from this paper.

We would like to avoid drawing conclusions about possible causal relationships between vegetation and megafauna because we do not have any extra information compared to Willerslev et al. 2014. At the same time we completely agree that megafauna extinction had/has significant effects on ecosystem function and hence plant community composition and that the significance of this has to be mentioned in the manuscript. We revised this section and added more references, though we cannot use all of those suggested due to space constraints.

293 I'm not entirely convinced this isn't an artifact of binning – the climatic periods in this analysis include periods of climate variability in some cases.

We rewrote this section. We also note the temporally uneven coverage on L 315. We have added the point that this limited our power to detect short-term variation (L 316-317).

321 Beringia?

We do not mean Beringia, but the wider area.

327 Please include these ages in a supplemental table.

Ages are given in the supplemental table of Willerslev et al. 2014. We aimed to avoid too much overlap with the earlier paper. Because of that, we refer to the published raw data but do not present it again. However, if the editor agrees that including this information once more in a supplemental table is justified, we would be happy to do that.

332 MIS 2 is 29 to 15 ka BP, and isn't necessarily synonymous with "LGM," which differs by region and represents a specific point at which glaciers reached their maximum extent. Similarly, "pre-LGM" is not synonymous with MIS 3, and your window (~60 to 25 ka BP) includes late MIS 4 and MIS 3. I'm not trying to nitpick here so much as to say that these time bins need to be clearer if they are to be meaningful. If you're using the terminology of defined stratigraphic markers, it's important to be precise.

Thank you. We have rewritten this section with greater precision.

Reviewer #6 (Remarks to the Author) (replacement for reviewer #1):

[Note from the editor: Reviewer #6 confidentially concluded that, overall, the responses to comments by reviewer #1 were satisfactory.]

My major comment is that there is a third alternative interpretation of the main findings, in addition to disappearing of megaherbivores, and "potential rupturing of mutualism" (which is a quite obscure mechanism, and could benefit from some explanation). It is now well-known that plant species respond individualistically to climate change, and that what we see as integrated plant communities is merely a mental construction by us, relatively short-lived humans. Thus, it may seem as plant communities during the late Pleistocene have been 'disrupted', while in fact everything represents a continuous process of change affecting individual species (due to climate, interactions with other

organisms, pure chance events etc). The pattern detected in this study may represent a similar phenomenon. Different plant species (with different mycorrhiza) respond to the same factors as today, but also to other still unknown factors, and over the time period covered (which has been sliced into three distinct periods) this appear as a 'rupturing' of the mycorrhizal communities.

We thank the referee for this very interesting perspective. We agree on several points. First, the 'rupturing of mutualism' point was indeed poorly presented, so we have rewritten this section (L 281-294). Second, the referee makes the point that species are known to have responded individualistically to climate change. Although we fully agree with this, our earlier text indeed did not make this explicit, so we have added this point to the Discussion (L 282). We believe that this section is now considerably clearer, and the two points - 'mutualist loss' and 'individualistic responses' are consistent and complementary. Regarding the study decision to assign samples to discrete time intervals, we understand the reviewer's concerns. Our approach summarised changes in relation to broad climatic periods, while in reality individual species may have followed continuous trajectories that reflected a number of drivers. Indeed our results (Table 2) indicate that communities differed considerably within climatic periods, as well as between periods, i.e., as the reviewer suggests there is plenty of scope for alternative drivers continually influencing individual species. However our approach, and that of Willerslev et al, was tailored to the available data: we have added a point to the Discussion acknowledging that the uneven spatial and temporal sample coverage limited our power to detect short-term or regional variations (L 317-318). On the final, related, point - that binning samples into climatic periods may have given the impression of 'rupturing' of mycorrhizal communities - we are less clear what the reviewer means. We did not measure mycorrhizal communities and did not intend to present our results as indicating rupturing of mycorrhizal communities. As noted above, we previously referred to rupturing of mutualisms as a possible explanation for the post-LGM patterns in plant mutualist trait abundance. We have rewritten this section, and hopefully clarified that we were speculating about a (hypothetical) process of spatial decoupling (i.e. the rupturing) between mutualist partners. We do not see why this interpretation is dependent on the sample binning approach, but we would be happy to be corrected if we have misunderstood.

Minor comments

Since pollination is included in the results and discussion, it should also be mentioned in the Abstract.

We now mention it the abstract.

I read the Abstract as a suggestion that the disappearance of megaherbivores may be a cause behind the changing community-level distribution of mycorrhizal types. In Willerslev et al. (2014) one gets the impression that the authors suggest it to be the other way around: the megaherbivores may have declined due to changing plant communities. In the text the authors make a remark on what (I interpret) they actually mean: that there is a positive feedback between herbaceous communities and megaherbivores, i.e. no simple cause and effect. This could be more clearly expressed in the Abstract.

We have tried to improve the Abstract, although we are strongly space-constrained.

On line 160, it is stated: “non-confounding mutualist traits”. Can you please clarify what you mean by this.

We apologise for the lack of clarity. We meant variation explained by a given mutualist trait after the variation attributable to other traits had been accounted for. For the comparisons in question (models including all non-confounding mutualist traits), we did not include traits together in the model if the categorizations were (partially) confounded: i.e., mycorrhizal status and type. We revised the text in order to make this clearer.

REVIEWERS' COMMENTS:

Reviewer #6 (Remarks to the Author):

I have no further comments. The authors have responded to my remarks in a satisfactory way.

Reviewer #7 (Remarks to the Author):

I fully agree with previous reviewer when they say that this paper shows a great way to leverage in a new and exciting way an incredible dataset.

I have come to revised this (3rd?) version, and consider that this revision was thorough in addressing the critical points raised by the reviewer –all of which I entirely agree. The main point of criticisms I have is the lack of a clear hypothesis tying why (1) the representation of mycorrhizal associations, (2) Bacterial nitrogen-fixing species prevalence, and (3) Pollinator relationships) should change as assemblages go from a pre-LGM to a post-LGM. This hypothesis could be framed in the context of latitudinal or temperature gradients – as they use the dissimilarity between their observations with current patterns seen along contemporary latitudinal and elevational gradients (L233-234).

Framing the paper in this way will make it easy to establish a clear connection between the reported changes, the reason they might occur, and the possible consequences of these changes as drivers of historic vegetation change.

Minor points:

L26-27 Please clarify what do you mean by “biogeographical reorganisation”. In a sensu stricto, this implies that biogeographical regions are reshuffled – which I do not think is the case. I guess you mean range changes that lead to changes in biogeographical patterns (species turnover for example).

L32: Provide a couple of examples of what you consider to be plant growth traits

L91-93: I would add one line after this section saying how these shifts relate to the idea that changes in the associated mutualist can affect shape the historic vegetation structure.

L100-101: please clarify why having an association can indicate the reliance on the symbiosis

L103: Based on the reviewer comments I would say "benefit" instead of "efficiency."

L107-112: Ok, my question now is: how having this association can affect shape the historic vegetation structure?

L115-124: After reading this part and the rest of the text I agree with the previous reviewer assessment that “The inclusion of plant-pollinator relationships is tenuous”. I don't see the added value of having this analysis here unless you set them up in the context of a hypothesis. If you are just describing the representation of pollination types on each period, I don't see what this is adding to the study.

L148-149: It would be appropriate to reference table 1 in this section

L183-185: Does your measurement of Niche positioning integrating across all the time periods to estimate the Mean species env position and the "Study" mean position?

L211-212: taxa that form "associations with" arbuscular mycorrhiza (AM)...

L238-240: based on this statement, I am wondering if the modern analogue gradient to be comparing the paleo gradient is a precipitation gradient not latitudinal and elevational gradient.

L283: by "the extinction of certain ecosystem engineers", you mean the loss of megaherbivores? Whatever the case if you need to add a reference here to support this statement.

L325-327: I am wondering if this is because you have a precipitation story in the pre_LGM-LGM-post_LGM sequence, while the evaluated contemporary patterns is one driven by temperature/latitude

Response to reviewers

Please find below the detailed responses (**formatted in bold**) to all referee comments.

Reviewer #6 I have no further comments. The authors have responded to my remarks in a satisfactory way.

Response: We are grateful for this response and appreciate the helpful suggestions made earlier by the referee.

Reviewer #7 (Remarks to the Author):

I fully agree with previous reviewer when they say that this paper shows a great way to leverage in a new and exciting way an incredible dataset.

I have come to revised this (3rd?) version, and consider that this revision was thorough in addressing the critical points raised by the reviewer –all of which I entirely agree. The main point of criticisms I have le lack of a clear hypothesis tying why (1) the representation of mycorrhizal associations, (2) Bacterial nitrogen-fixing species prevalence, and (Pollinator relationships) should change species as assemblages go from a pre-LGM to a post-LGM. This hypothesis could be framed in the context of latitudinal or temperature gradients – as they use the dissimilarity between they observations with current patterns seen along contemporary latitudinal and elevational gradients (L233-234).

Framing the paper in this way will make it easy to stablish a clear connection between the reported changes, the reason they might occur, and the possible consequences of these changes as drivers of historic vegetation change.

Response: We are grateful for this comment. We have revised the manuscript and have attempted to improve the flow of the text throughout. We have also introduced a hypothesis regarding our expectation about changes in the dominance of mutualistic types through time. We note that this hypothesis was included in earlier versions and is one that is necessarily fairly general, reflecting the novelty and descriptive nature of the study.

Minor points:

L26-27 Please clarify what do you mean by “biogeographical reorganisation”. In a sensu stricto, this implies that biogeographical regions are reshuffled – which I do not think is the case. I guess you mean range changes that lead to changes in biogeographical patens (species turnover for example).

Response: We accept that this was a misleading phrase. We actually meant to describe the results of what might be broadly described as biogeographic processes. The example we mention in the abstract is spatial decoupling of potential mutualistic partners, which might result from the sort of range changes the reviewer describes; but we also consider extinction of megaherbivores to fall into this broad category. We have changed the text to reflect this more clearly.

L32: Provide a couple of examples of what you consider to be plant growth traits

Response: Growth rate serves as an example of a growth trait. However, we actually meant life-history traits, which is a broader category including various other traits (e.g. seed mass, life span, shade tolerance etc.). We have changed the text to reflect this.

L91-93: I would add one line after this section saying how these shifts relate to the idea that changes in the associated mutualist can affect shape the historic vegetation structure.

Response: We have added an explanatory sentence: One might expect that temporal change in the predominance of mycorrhizal types should follow a broadly analogous pattern to those observed in relation to latitude and altitude, e.g. such that the predominance of mycorrhizal types in historically cooler periods most closely resembles the pattern in contemporary high latitude vegetation.

L100-101: please clarify why having an association can indicate the reliance on the symbiosis

Response: Mycorrhizal status – being obligate or facultative with respect to forming the association – is, by definition, a measure of plant reliance on mycorrhiza (another measure that is used less frequently is the mycorrhizal growth response, sometimes also called mycorrhizal dependency). We improved the wording in order to provide more clarity.

L103: Based on the reviewer comments I would say "benefit" instead of "efficiency."

Response: We agree and have replaced the word.

L107-112: Ok, my question now is: how having this association can affect shape the historic vegetation structure?

Response: We agree that this aspect was not fully explained in the text. We added a sentence about how we expected the share of N-fixing plant species to change along a climatic gradient. N-fixing plants can shape entire communities by improving N supply and hence changing the competitive balance between plant species (the potential importance of N fixers is also noted in this section).

L115-124: After reading this part and the rest of the text I agree with the previews reviewer assessment that "The inclusion of plant-pollinator relationships is tenuous". I don't see the added value of having this analysis here unless you set them up in the context of a hypothesis. If you are just describing the representation of pollination types on each period, I don't see what this is adding to the study.

Response: We considered this issue in response to an earlier comment by referee 5. We agree that the categorisation of pollination types that was possible was not as detailed as we would have liked. We also note that the categories we defined did not exhibit a marked trend in the data. The only reason

we would like to retain the pollination data is because we believe the results will nonetheless be of interest to paleoecologists. Most analysis of historical vegetation is based on pollen, and for this reason, understanding how the relative abundance of wind pollinators – the predominating pollen producers and a category we could measure – has changed in time is important. We have thus kept the analysis, but we would accept removing it if the Editor feels strongly it should be removed.

L148-149: It would be appropriate to reference table 1 in this section

Response: We agree and added a reference to Table 1.

L183-185: Does your measurement of Niche positioning integrating across all the time periods to estimate the Mean species env position and the "Study" mean position?

Response: This is correct. We have added this point to the text in parentheses

L211-212: taxa that form “associations with” arbuscular mycorrhiza (AM)...

Response: Plant species associate with AM fungi in order to form arbuscular mycorrhiza (mycorrhiza means ‘fungus-root’). Hence we might say either ‘...taxa that form associations with AM fungi’ or ‘taxa that form arbuscular mycorrhiza’. We would prefer to use the second version because referring to association with fungi may lead the reader to expect that fungal taxa will be directly addressed.

L238-240: based on this statement, I am wondering if the modern analogue gradient to be comparing the paleo gradient is a precipitation gradient not latitudinal and elevational gradient.

Response: We agree that precipitation may be an important driver of changes in the dominance of different mutualisms through time. Alongside temperature, it is an important co-driver of regional variation in mycorrhizal types (Bueno et al. 2017 Global Ecol Biog). However, as we indicate in the Discussion, there are a number of other possible drivers (temperature, CO₂, soil resources, herbivory, spatial decoupling of mutualistic partners). Because we have no data to disentangle the relative effects of these various drivers, we would rather not speculate about the possible effect sizes of different factors.

L283: by “the extinction of certain ecosystem engineers”, you mean the loss of megaherbivores? Whatever the case if you need to add a reference here to support this statement.

Response: We indeed meant megaherbivores and state this more clearly in the current version (including a reference).

L325-327: I am wondering if this is because you have a precipitation story in the pre_LGM-LGM-post_LGM sequence, while the evaluated contemporary patterns is one driven by temperature/latitude

Response: We agree that precipitation is potentially a major driver. However, we cannot state this with any certainty because we have no empirical data to estimate the possible effect sizes of other factors, such as soil P and N, megaherbivore extinction, CO2 and spatial decoupling.